# Unraveling the Complexities of Beef Marination: Effect of Marinating Time, Marination Treatments, and Breed

**DOI:** 10.3390/foods13182979

**Published:** 2024-09-20

**Authors:** Sena Ardicli, Ozge Ardicli, Hakan Ustuner

**Affiliations:** 1Department of Genetics, Faculty of Veterinary Medicine, Bursa Uludag University, Gorukle District, Bursa 16059, Türkiye; 2Milk and Dairy Products Technology Program, Division of Food Processing, Karacabey Vocational School, Bursa Uludag University, Bursa 16700, Türkiye; 3Department of Animal Science, Faculty of Veterinary Medicine, Bursa Uludag University, Gorukle District, Bursa 16059, Türkiye; hustuner@uludag.edu.tr

**Keywords:** beef, marination, meat quality, texture, consumer preference

## Abstract

The present study focused on evaluating the effects of beef marination on quality traits and consumer acceptability. In this context, *m. longissimus thoracis et lumborum* and *m. semimembranosus* samples (n = 192) were obtained from Aberdeen Angus, Hereford, Charolais, and Limousine bulls and were marinated with milk (pasteurized, 100%), garlic and olive oil (2.35 g/500 mL), and lemon (citrus) juice (31% orange juice, 31% lemon juice, 38% distilled water) for 12, 24, and 72 h. Marinade components were selected based on traditional culinary practices and their scientifically proven effects on meat quality. Beef samples on day 0 and non-marinated samples were used as control groups. Beef color, water holding capacity, pH, cooking loss, and Warner–Bratzler shear force were measured three times for each sample. A taste panel assessment was performed to determine the sensory characteristics. Statistical analysis was performed using general linear model (GLM) procedures followed by Tukey’s post-hoc comparison. Results revealed that marination time, as well as its two- and three-way interactions, significantly influenced beef quality parameters. These results indicate that the cattle breed is an important factor in evaluating the effectiveness of beef marination applications. The olive oil−garlic marinade was the most preferred by the panel across both types of muscle, as indicated by sensory evaluation results. The findings will not only enrich the scientific literature but also have practical implications for the beef industry.

## 1. Introduction

Meat quality has gradually become a focus of the meat industry. It should first be emphasized that the application of novel techniques to improve meat quality is highly associated with consumer preferences, because quality is defined by the parameters that the consumer perceives to be preferable [1]. Many methods for improving the traits of meat quality have been developed, including various marination techniques, such as injection of marinades, subjecting the meat to pressure, and the use of different marinade solutions containing lemon juice, salt, milk, spices, and curds. [2,3,4,5].

The primary objective of marination is to provide a variety of meat products with high sensory acceptability [5]. Although marination is one of the most popular applications in the meat marketplace, especially for upscale restaurants or specialty shops, commercial marinades commonly have limited influence on meat tenderness, which is known to be one of the most important and decisive parameters in the meat industry. However, the marination process enhances or complements meat flavor, which results in an improvement in palatability [2]. On the other hand, marination may have other important effects, such as decreasing microbial growth or maintaining the desired meat color for longer periods [3]. It should be noted that some applications, such as marination in calcium chloride, may result in various undesired situations, including abnormal flavors and odors [6,7] or faster discoloration [8], even if they improve tenderness [3,6]. The influence of the marination process on the quality parameters of meat is complex, and the mechanisms underlying the alterations in meat quality caused by marinades involve several factors. For instance, the rates of conversion of collagen to gelatin at low pH during cooking or proteolysis caused by cathepsins, structural modifications caused by swelling, and the chemical properties of marinades may induce alterations in tenderizing action [2,9]. A similar interpretation may apply to meat color. Marination significantly influences meat color, primarily through the interaction of marinade acids with meat pigments such as myoglobin [6]. For instance, marinades that contain citric acid can significantly reduce the meat’s pH. This reduction in pH causes a lightening of meat color. Marinades containing high amounts of garlic produce a dark color. It is important to note that sugars in garlic can affect color via the Maillard reaction, which enhances browning [10]. The acidic environment promotes this change by altering the state of myoglobin, the protein responsible for meat color, resulting in a paler appearance. These biochemical interactions are critical for enhancing the visual appeal of marinated meat, contributing to consumer preference and market value [2,6,11].

In choosing the components of the marinade for this study—milk, olive oil, garlic, and lemon juice—we considered both traditional culinary practices and their scientifically documented impacts on meat quality. Dairy products were used for their tenderizing effect due to the presence of calcium and lactic acid, which activate enzymes that break down proteins [12]. Olive oil was selected for its fat content, which enhances the infusion of flavor and moisture retention in meat [13]. Garlic and lemon juice are known for their strong antimicrobial properties and ability to enhance meat flavor and tenderness through acidity, which aids in the breakdown of meat fibers [2,14,15,16]. Previous research combining these natural marinades has demonstrated enhancements in meat safety, including microbial quality and fat oxidation, as well as improvements in physical characteristics such as color and texture, as determined through instrumental measurements [12].

The beef industry has focused on developing strategies to boost the market value and customer satisfaction associated with commercial meat products. Among these strategies, beef marination stands out as a particularly effective technique [2,12]. Its growing popularity in the beef industry has opened substantial marketing opportunities driven by increasing consumer demand for marinated beef, particularly in high-end dining establishments and specialty stores. This trend reflects a broader consumer preference for ready-to-cook products that deliver enhanced taste and convenience.

Previous studies on beef marination have focused primarily on choosing an adequate marinade agent and/or concentration, but information about alterations in the quality parameters of meat during marination for varied lengths of time (especially long-term marination, >24 h) is limited. Moreover, there is a lack of knowledge regarding the effects of breed on marinated beef. Therefore, the main objective of this study was to perform a comprehensive analysis of the marination of beef and, accordingly, to determine the influence of marinating treatments (milk, olive oil, garlic, and lemon juice), marinating time, and breed on the quality parameters of meat, including color, water holding capacity (WHC), pH, cook loss (CL), shear force, and sensory traits.

## 2. Materials and Methods

### 2.1. Animals

Four beef cattle breeds, Aberdeen Angus, Hereford, Charolais, and Limousine (n = 48; 12 from each breed; all bulls), were used in the present study. All animals were fed ad libitum with the same diets and were bred on the same commercial farm located in the South Marmara region of Türkiye (40°15′28.04″ N, 29°30′50.18″ E) according to the same management practices. The feed mixture for this formulation consisted of a variety of components aimed at providing a balanced diet. Corn silage and wheat straw formed the bulk of the mixture, contributing 28% and 27%, respectively. This was followed by macaroni pellets at 12%, corn bran at 11%, and corn gluten feed at 10.5%. Sunflower meal added another 6.5% to enhance the nutritional profile. To ensure adequate vitamin and mineral intake, a dedicated supplement was included at 5% of the total composition. All animals were fed for up to 24 h before slaughter and then fasted to ensure consistent gastrointestinal conditions, adhering to standard pre-slaughter protocols. The total herd size was 1200 cattle. At the end of the fattening period, animals of the same age (466.30 ± 3.60 d) and final weight (605.25 ± 1.25 kg) were selected for subsequent evaluations. The duration of transportation from the farm to the slaughterhouse was approximately 1 h, and the animals were slaughtered by means of exsanguination. The resting time of the animals in the slaughterhouse pens before slaughter was approximately 8 h. All animals were dressed using standard commercial procedures. All carcasses were electrically stimulated for 30 s at 60 volts, suspended by the Achilles tendons, and chilled for 24 h at 4 °C in a ventilated room. It is important to note that particular attention was paid to ensuring that all farm and slaughterhouse conditions were exactly the same for all animals. As a result, the selected animals were uniformly exposed to identical environmental factors, thereby constituting a population in which breed was the sole variable.

### 2.2. Muscle Samples, Treatments, and Experimental Design

Before the experimental design, a preliminary evaluation was performed to select the appropriate marination treatments for the main study. In this context, apart from studies on marinating beef in the literature, explanatory information provided by chefs working in upscale restaurants (meat-specific) was used in the present study.

*M. longissimus thoracis et lumborum* (n = 96) and *m. semimembranosus* (topside) (n = 96) were dissected from each carcass and trimmed of visible external fat (Figure 1). Postmortem processing, including deboning, was conducted 24 h after slaughter to standardize the effects of *rigor mortis* on the assessments of meat quality. Samples were cut transversely at the midlength to provide four equal sections. Initially, analyses of meat quality, including color, WHC, cooking loss, and shear force, were performed on fresh samples to determine the corresponding values on day 0. The remaining sections were prepared for the marination process. Beef strips measuring 40 × 35 × 5 mm were cut parallel to the direction of the muscle fibers using a custom-made template (35 × 40 mm) with a uniform thickness of 5 mm, as described by Burke et al. [2]. Muscle samples were allocated to the following marination treatments: (I) non-marinated control, (II) milk marination, (III) olive oil and garlic marination, and (IV) lemon juice marination. The reagents for the marinade solutions were purchased from a local supermarket. All marinades were prepared on the same day and maintained at 4 °C until required. Pasteurized whole milk (100%, pH ~6.6) was used for the milk marinade, while the lemon (citrus) juice marinade was prepared according to Burke et al. [2]. In this context, a citrus juice marinade (designated as lemon marinade in this study), consisting of 31% freshly squeezed lemon and orange juice from fruit purchased in a local supermarket, was prepared in distilled water (38%). The olive oil and garlic marinade consisted of 2.35 g garlic and 500 mL olive oil (pH = 5.38), which were mixed homogeneously. The marinating process involved immersing slices of muscle in marinades to ensure complete coverage of the meat samples. The ratio between meat and marinade was fixed at 1:10 [12]. The individual muscle samples, including the non-marinated control samples, were packed in high-density polyethylene (HDPE) refrigerator storage containers and wrapped with commercial-grade polyvinyl chloride film (Sera, Manisa, Türkiye). The beef samples were marinated for 12, 24, and 72 h at 4 °C in a refrigerator (Arçelik 554210 MB No Frost refrigerator, Istanbul, Türkiye) to determine the effects of marination time on meat quality parameters. Samples were turned approximately every 6 h. Concurrently, the pH of the marinades was measured and the marinades were replaced with fresh ones to maintain consistent pH levels throughout the marination process.

For all quality parameters of beef, three repeated measurements were performed, and the average was evaluated as the final value for each sample. It is important to note that concerning the marination treatments, the excess marinade was removed before the determination of the quality parameters of the meat, and samples were blotted with a paper towel to prevent mismeasurements resulting from the marinades.

### 2.3. Evaluation of Meat Color

Color measurements were performed using a Minolta CM508d reflectance colorimeter (Konica–Minolta Inc., Ramsey, NJ, USA) with a glass plate covering the measuring aperture (8 mm diameter) for standard observation at 10° with the D65 illuminant. The device was calibrated against a white plate provided by the manufacturer according to the directions of the instrument manual. It was set to make three measurements, and their average was taken. Following the removal of the marination solutions by applying paper towels lightly to the sample surfaces, triplicate color measurements were performed on each sample on the cut surface of the fat-free area. The average of these measurements was used as the final value. The color was expressed in terms of the CIE *L**, *a**, and *b** color coordinates. Moreover, the chroma value (*C**) and hue angle (*h*°) were calculated according to the formulas indicated by Węglarz [17], as follows:$$C*=\sqrt{{\left(a*\right)}^{2}+{\left(b*\right)}^{2}}$$
$$h{}^{\circ}=\arctan\left[\frac{b}{a}\right]\;$$

### 2.4. WHC

WHC (%) was estimated by ascertaining the expressible moisture content of the sample using a modified Grau and Hamm method [18]. In this context, a meat sample weighing 5 g was placed on 10 cm diameter filter papers between two petri plates and pressed under 2.250 kg for 5 min. Following the removal of the filter papers and the weight, WHC, as a percentage, was calculated as (final filter weight − initial filter weight)/meat sample weight × 100 [19].

### 2.5. pH Measurement

The sample pH was measured using a digital pH meter (Testo 205; Lenzkirch, Germany). Before the measurements, a calibration procedure was applied to the device with pH 4.01 and pH 7.00 standard buffer solutions (Testo; Lenzkirch, Germany) as instructed by the manufacturer. The measurements were repeated three-times at random points in each sample. Carcass pH (designated as the 0 h control) was measured in the longissimus thoracis muscle between the 12th and 13th ribs 1 h postmortem.

### 2.6. CL

To determine CL, meat samples were first weighed using an analytical balance (Radwag AS220/C/2, capacity 220 g, readability 0.10 mg, Bracka, Kraków, Poland) to ascertain the value of the initial sample weight (raw weight). All samples, each 150 ± 2.80 g, were uniformly cut to the same shape. Afterward, the samples were cooked in a 75 °C water bath (Nuve BM 302, Ankara, Türkiye) for 1 h and cooled in running tap water for 1 h. The samples were then removed from their packages, blotted with a paper towel to remove excess moisture, and weighed again to determine the cooked sample weight. CL was estimated as follows: (weight before cooking − weight after cooking)/weight before cooking × 100. Cooked samples were also used to determine the shear force value [19].

### 2.7. Warner−Bratzler Shear Force Assessment (WBSF)

Following the corresponding marination time (12, 24, and 72h), both marinated and control samples were cooked and allowed to cool as previously described. Three cylindrical cores parallel to the longitudinal muscle fiber orientation with approx. 10 × 10 mm surface area and min. 25 mm length was removed. Shearing was accomplished with a Universal Testing Machine (Zwick/Roell Z0.5, Ulm, Germany) equipped with a V-shaped Warner–Bratzler attachment (60° triangular aperture). The full-scale load was 50 kg, and the crosshead speed was 150 mm/min. The shear force values for at least three cores (3–6) were determined, and the average of these measurements was accepted as the final value for each sample [19].

### 2.8. Sensory Analysis

The panelists (n = 28) were recruited from among the academicians of the Bursa Uludag University Faculty of Veterinary Medicine. Some of the panelists had participated in previous sensory evaluation studies. Before the sensory assessment, the samples were wrapped individually in aluminum foil, cooked using a pre-heated double-plated electric oven (Nuve FN 120, Ankara, Türkiye) at 200 °C, and labeled with three-digit numbers. The steaks were individually arranged on mesh racks set within aluminum pans, with two thermocouples embedded at the geometric center of each to monitor the internal temperature. The cooking process lasted approximately 20 min and was continued until the samples reached a final internal temperature of 75 °C. Following the removal of cover fat and connective tissue, the samples were cut into 20 × 20 × 10 mm sub-samples and served individually on plastic plates. The panelists were instructed to rinse their mouths with water and smell ground coffee before the sensory evaluation and between subsequent tastings. The sample’s presentation order was arranged as a five-member group of panelists, and the evaluation was accomplished using an 8-point hedonic scale. In this respect, the assessment was based on seven criteria, namely odor, flavor, tenderness, juiciness, color, general acceptance (absence of abnormal flavor or odor), and overall liking, as follows: odor (8 = like extremely, 1 = dislike extremely); beef flavor (8 = extremely intense, 1 = extremely bland or no flavor); intensity of tenderness (8 = extremely tender, 1 = extremely tough); level of juiciness (8 = extremely juicy, 1 = extremely dry); beef color (8 = like extremely, 1 = dislike extremely); general acceptance (8 = extremely desirable, 1= extremely undesirable, abnormal flavor or odor); and overall like (8 = like extremely, very definitely would purchase, 1 = dislike extremely, very definitely would not purchase). Each panelist received at least three cubes of each sample from the corresponding sub-groups (beef from four breeds, the day 0-control, and beef marinated with four different marinades for 12 h and 24 h). All sensory analyses were performed under the same environmental conditions and conducted by the same panelists.

Sensory evaluation was conducted separately for the two muscle types across the four sessions. In the first two sessions, the panelists were presented with two sub-samples from each breed, totaling eight sub-samples per session. To prevent order effects and bias, the presentation of samples was randomized in each session. Each session lasted approximately 1 h, including brief breaks between samples to clean the palate and reset sensory receptors. This setup was specifically designed to prevent fatigue and ensure sharp sensory perception throughout the assessment.

### 2.9. Statistical Analysis

All statistical analyses were performed using SPSS version 23.0 (IBM, Armonk, NY, USA). The homogeneity of variance was checked using Levene’s test. Selected quality parameters of beef were assessed by analysis of variance (ANOVA) using general linear model (GLM) procedures based on the following model:$$Y_{ijklm}=\mu +M_{i}+B_{j}+X_{k}+I_{l}+e_{ijklm}$$
where *Y_ijklm_* = the studied traits; *μ* = the overall mean; *M_i_* = the fixed effect of marination treatment (*i* = non-marinated control, milk, olive oil-garlic marination, and lemon juice); *B_j_* = the fixed effect of breed (*j* = Aberdeen Angus, Hereford, Charolais, Limousine); *X_k_* = the fixed effect of marination time (*k* = 0 h, 12 h, 24 h, 72 h); *I_l_* = the fixed effect of the interactions; *e_ijklm_* = random error.

The model was simplified by the removal of non-significant fixed effect terms, including the side within the carcass, the portion within a side, and the sub-portion within the portion, as suggested by Toohey et al. [20]. The session factor was added as a random effect to account for potential variability between sessions. Since it was not significant, it was excluded from the final statistical model. The statistical analyses regarding *m. longissimus thoracis et lumborum* and *m. semimembranosus* (topside) were performed separately. Concerning the semimembranosus muscle, the data were available for the 0 h control and the 12 h and 24 h treatments. The GLM procedure was also used to evaluate sensory differences using the fixed effects of marination time (marination for 12 h and 24 h), marination treatment, and cattle breed. All possible two- and three-way interactions were tested. The least-squares means (LSMs) and standard errors of the mean (SEMs) were computed for all parameters, and differences between treatments were tested. The significance levels for each parameter were assessed at *p* < 0.05, *p* < 0.01, and *p* < 0.001. Tukey’s post-hoc comparisons of means were conducted if significant differences were evident (Figure 1). Evaluations that were not confirmed by Tukey’s multiple comparison test are also included in the report.

## 3. Results and Discussion

Various studies have been conducted to enhance consumer acceptance and improve the quality parameters of beef, such as flavor, color, and tenderness. These studies provide insights aimed at increasing the retail value and consumer satisfaction of commercial meat products [4,21]. In this respect, marination of beef is one of the most popular methods, providing a large marketing opportunity in the beef industry because the demand for marinated products has been growing continuously, especially in upscale restaurants or specialty shops. Marinating beef is a complex process that is influenced by various chemical and biological factors. With the rising demand for top-notch dining experiences and uniquely marinated meat, this topic has garnered significant attention. It is crucial to scientifically evaluate different marination methods using quantitative analyses to characterize, optimize, and ensure the reproducibility of the outcomes. This emphasizes the need for comprehensive and meticulous research in this field. Moreover, to our knowledge, no comprehensive study in the existing literature has thoroughly examined the impacts of the breed of beef cattle and its interactions with quality parameters. In the current study, we comprehensively evaluated the characteristics of beef from *m. longissimus thoracis et lumborum* and *m. semimembranosus* (topside) samples obtained from Aberdeen Angus, Hereford, Charolais, and Limousin cattle based on milk, olive oil−garlic, and lemon juice marination treatments. In this work, an additional aim was to test the effects of some traditional native beef-marinating practices in the Aegean region of Türkiye on beef quality. Accordingly, we selected an olive oil-garlic marinade for our analysis. 

A summary of the statistical evaluation regarding the attributes of beef quality in *m. longissimus thoracis et lumborum* is presented in Table 1, while the results for *m. semimembranosus* (topside) are presented in Table 2. Regarding the two-way interaction analyses, the least-squares means and significance levels for all factors are presented in Appendix A. The differences in the instrumental and sensory properties between the treated samples obtained from *m. longissimus thoracis et lumborum* and *m. semimembranosus* (topside) are discussed separately for each trait.

### 3.1. Beef Color and pH

As expected, time was effective on all attributes of beef color in both muscles at different levels of significance. When considering the mean values of the color parameters of *m. longissimus thoracis et lumborum*, a notable decreasing trend was observed in the *a**, *b**, and *C** values (Figure 2a). Specifically, after 72 h of marination, the beef exhibited significantly lower values of *a** (6.84 ± 0.42) and *b** (12.23 ± 0.31), as detailed in Table 3. Concerning *m. semimembranosus*, marination time had a significant effect on all color parameters, excluding *b** (Table 2).

In *m. semimembranosus*, a significant decrease was observed in the *a** and *C** values (*p* < 0.001). While no significant differences were noted between 0 and 12 h, a notable decrease was particularly evident at 24 h of marination (Figure 3a). It is also apparent from the data presented in Table 3 and Table 4 that the semimembranosus muscle exhibited darker coloration than *m. longissimus thoracis et lumborum* after 24 h of marination. In both muscles, an increase in marination time led to an increase in the *L** value, resulting in brighter beef following the marination process. Numerous traits contribute to the quality of meat [1,17,22]. However, one of the most critical criteria that consumers use to assess the value of meat in relation to its price, as well as its health properties, is color [23,24,25]. Meat color directly influences the consumer’s first impression and often dictates their purchasing decisions. Furthermore, consistent meat color is associated with freshness and quality, impacting consumer’s trust and satisfaction. As such, understanding and controlling variations in meat color are essential for meat producers to meet consumer expectations and enhance marketability.

In this study, our findings indicated that marination time significantly affected the color parameters of beef, a trend that was corroborated by panel evaluations of *m. longissimus thoracis et lumborum* (*p* < 0.05) and *m. semimembranosus* (*p* < 0.001). A notable difference was observed between the 12 h and 24 h marination periods, with panelists expressing a preference for meat marinated for 24 h for both muscle types (Table 5 and Table 6). As our study has effectively demonstrated, the preference for longer marination times is significant. This underscores the necessary duration for marination solutions to fully penetrate the meat and influence its characteristics. However, it is important to note that marination solutions with a low pH, such as those containing lemon, may initiate a precooking process in the meat, thereby altering its color. Panel evaluations confirmed that a 24 h marination period is more effective in achieving the desired natural and cooked appearance of the meat.

The correlations between meat pH and all color parameters were found to be significant. For example, an undesired increase in meat pH may lead to deterioration of all color parameters [17]. This trend in color is consistent with the well-established relationship between beef color and pH, where changes in pH directly affect beef coloration [24,26]. The pH value of beef is crucial for its quality, influencing color stability and microbial growth [27]. Additionally, protein biomarkers during the early postmortem period can predict subsequent declines in pH and the development of color in beef [28]. Beef with a high pH, characterized by elevated raw muscle pH, can protect myoglobin from heat denaturation. This protection results in a persistently red or pink appearance, necessitating higher cooking temperatures to achieve the desired cooked appearance [29] and color stability [30]. In this study, the impact of marination time on pH was found to be significant. The pH of the beef initially decreased, reaching its lowest point during the first 12 h from the baseline (control samples at 0 h), and subsequently began to increase in both muscle types (Figure 2b and Figure 3b). By the end of the 72 h marination period, the pH value had risen to 5.18 in *m. longissimus thoracis et lumborum* (Table 3). The 24 h marination led to a mean pH of 5.42 ± 0.09 (Table 4). These alterations in pH are critical, as they contribute to the color changes observed in meat, which can influence consumers’ perception and acceptance of meat products. Beef pH patterns result from a complex and dynamic biochemical transformation process that begins at slaughter and must be carefully examined at each stage [1,17,24].

In this study, the four different marination treatments we implemented resulted in significant changes in color properties. While these treatments significantly affected all parameters except the *b** value in *m. longissimus thoracis et lumborum* (Table 1), interestingly, they only influenced the *L** value in *m. semimembranosus* (Table 2; this effect was also not confirmed by the post-hoc analysis). This outcome demonstrates that the same marination technique can produce varying results across different muscles. In *longissimus thoracis et lumborum muscle*, the highest *L** and *h** values were observed for the milk marinade (Table 3). It appears that marination generally decreases the redness of beef, as indicated by the *a** value; the highest *a** values were found in non-marinated samples. As shown in Table 3, the *b** value did not increase significantly after marination, which can be considered a positive outcome in terms of reducing the risk of bacterial growth. Surprisingly, marination treatment did not cause significant changes in *m. semimembranosus*. Similar means were observed across the four different marination treatments (Table 4).

In both muscle types, the marination treatments resulted in significant changes in pH, as shown in Table 1 and Table 2. As expected, the lowest pH values were observed for the lemon juice marinade (Table 3 and Table 4). Furthermore, in *m. longissimus thoracis et lumborum*, the olive oil-garlic marinade induced a more pronounced decrease in pH compared with the semimembranosus muscle. These variations in pH can be attributed to the differing fat and protein content of each muscle type. Acidic marination involves immersion of meat in an acidic solution. In this context, lemon juice, a common ingredient in marinade, contains citric acid that influences the pH of the marinade solution [2]. These acidic solutions have traditionally been used to soften and flavor meat through marination [11]. Thus, the pH of the marinade can significantly impact the textural properties of meat, with both high and low pH levels altering the texture of the meat, albeit at different rates [31]. Moreover, marination can influence the fatty acid composition of meat, with the antioxidant activity in marinades potentially suppressing the oxidation of certain fatty acids [32]. An acidic environment enhances proteolysis, primarily through the activation of cathepsins, which are enzymes responsible for breaking down proteins. Additionally, acidic conditions facilitate the conversion of collagen, a robust protein in connective tissues, into gelatin during the cooking process [2].

Marinating is recognized as a means of enhancing the quality and versatility of meat cuts. Commercial marinades improve palatability by enhancing or complementing the flavor of meat [2,33]. Flavor perceptions can vary significantly, with the degree of cooking and aroma quality varying across different countries. In this study, marination not only altered the flavor and odor, but also significantly affected (*p* < 0.001) the perception of beef color by the panel (Table 1 and Table 2). In this context, the olive oil-garlic marinade received the highest scores for both *m. longissimus thoracis et lumborum* (Table 5) and *m. semimembranosus* in the panel (Table 6). The fact that this marination treatment resulted in higher color panel scores than the non-marinated samples in both muscles underscores the critical role that marination can play. This study demonstrated the importance of effective marination, particularly with olive oil and garlic, through improvements in both meat color indicators and panel evaluations.

Cattle have diverged from their ancestral stock into breeds with distinct phenotypic differences due to many years of artificial selection. Subsequent intensive selection tailored to specific purposes has led to the establishment of specialized beef and dairy cattle breeds, each exhibiting significant anatomical, physiological, and genetic distinctions [34]. Consequently, it is a reasonable hypothesis that these breeds would respond differently to various marination ingredients owing to the unique structure and composition of their muscles. Most marination research tends to overlook the breed factor, often beginning analyses with retail cuts. In this study, we utilized meat samples from animals of known breeds, allowing us to explore the effects of marination not only in terms of time and treatment, but also in relation to breed variation. As shown in Table 1, the breed did not individually influence color traits in the *m. longissimus thoracis et lumborum*. However, the *b** and *C** values, as well as the color assessments from the panel, were significantly affected by the breed factor (*p* < 0.05). Specifically, samples from Hereford cattle exhibited higher *b** and *C** values (Table 4). Additionally, samples of the semimembranosus muscle from Aberdeen Angus and Limousine breeds received significantly higher scores in the panel assessments (Table 6).

Two-way interactions facilitate the evaluation of the combined significance of multiple factors, provided that a sufficient sample size is available. This approach enables more effective judgments compared to the assessment of individual effects alone. In this study, two-way interactions of all factors were examined in detail (Appendix A). Three-way interactions were also thoroughly evaluated (Table 7, Table 8, Table 9, Table 10 and Table 11). Concerning beef color, significant differences were observed due to the interaction between marinating time and marination treatment, as documented in Table 1. Specifically, marination in milk for 72 h resulted in the highest *L** and *a** values in the *m. longissimus thoracis et lumborum*, as detailed in Appendix A. In contrast, the non-marinated samples exhibited the highest *b** and *C** values. Notably, in the panel assessment, Aberdeen Angus beef marinated with olive oil and garlic achieved the highest mean color score (6.14 ± 0.23) across both muscle types, as shown in Appendix A. However, two-way interactions involving breed did not lead to significant changes in the color parameters of beef, as indicated in Appendix A. Furthermore, the time × treatment × breed interaction showed no significant effect on beef color traits (Table 1 and Table 2). We found that this three-way interaction had a significant effect on beef pH (*p* < 0.05). Non-marinated (0 h control) samples had higher pH values than the experimental samples (Appendix A). Furthermore, Charolais beef marinated in either milk or olive oil-garlic for 72 h had relatively elevated levels of beef pH compared to other samples.

Quantitative evaluations of the effects of different marination techniques on beef pH and comparisons of these data with color characteristics provide valuable insights. In our study, the changes in pH and beef color, along with their correlations observed in panel evaluations, showed agreement, and more precisely, they confirmed each other.

### 3.2. WHC and CL

Marinating time had an effect on WHC and CL at different levels of significance in both muscle types (Table 1 and Table 2). In *m. longissimus thoracis et lumborum*, the WHC was similar in the 0 h control, 12 h, and 24 h samples, yet a sharp decrease was observed in the 72 h samples. The highest CL was recorded in the samples marinated for 24 h (35.87 ± 0.79). While the marination treatments did not cause a significant change in WHC, it was found that lemon marination resulted in the highest CL (35.89 ± 1.09), as indicated in Table 3. The breed did not produce a significant individual effect (*p* > 0.05). In the case of the semimembranosus muscle, the highest WHC values were observed in the 0 h control samples, while the samples marinated for 12 h and 24 h exhibited lower WHC levels (Table 4). Further, 12 h marination led to the highest CL (*p* < 0.01). Marination treatment significantly affected WHC in *m. semimembranosus* but not in the *m. longissimus thoracis et lumborum*. Remarkably, milk marination was characterized by the highest WHC (13.96 ± 0.52), as shown in Table 4. Concerning two-way interactions, time × treatment influenced the CL in the *m. longissimus thoracis et lumborum* (Table 1) and the WHC in the *m. semimembranosus* (Table 2). Lemon marination for 24 h led to the highest CL in the longissimus muscle (Appendix A), while the non-marinated 0 h control exhibited the highest WHC in *m. semimembranosus* (Appendix A). In this study, these findings confirmed that marination generally leads to a decrease in WHC. We defined a significant effect of the breed × marinating time × marination treatment interaction on WHC (*p* < 0.05) in *m. semimembranosus*. In this context, milk marination of Hereford beef samples for 24 h had the highest WHC (18.99 ± 1.96), while the 24 h olive oil-garlic marination of Limousine beef had the lowest values (4.63 ± 1.19), as shown in Table 10. WHC is a crucial attribute of beef samples as it directly influences many important properties, including the overall palatability of the meat [35,36]. The ability of meat to retain water during storage, processing, and cooking is essential for maintaining juiciness and preventing the dryness of the final product [37]. Beef with a high WHC retains more moisture during cooking, leading to juicier and more tender meat. This is typically more desirable for most cooking methods and cuts, as it enhances the eating experience and can also affect the weight of meat, which is crucial for pricing in commercial contexts [38]. It is also affected by many factors, including the pH of the marinade and type of beef muscle [39]. Conversely, beef with a low WHC loses more moisture during cooking, which can result in drier and tougher meat. However, for certain products, such as dried or cured meats, a lower WHC may be preferable for achieving the desired texture and concentration of flavors. Additionally, the composition of marinades, including essential oils, spices, and fruit juices, can significantly influence the WHC and overall quality of meat. For instance, the presence of essential oils in marinades has been demonstrated to enhance beef quality. This improvement is often accompanied by favorable evaluations from the sensory panels [13]. These ingredients not only modify the WHC but also contribute to the aromatic and flavor profile of the meat, further enhancing the consumer’s eating experience.

From a broader perspective, traits such as pH, WHC, CL, and color are interconnected, contributing to the complexity of beef quality. We propose that the breed factor should be considered when assessing beef quality, particularly in processes such as marination. This is because the structural characteristics and intramuscular fat content of different cattle breeds significantly influence this effect [40]. Marination has been linked to juicier meat and less water loss during cooking, which enhances both the texture and overall quality of the meat [14] and can influence the lipid oxidation level of beef [41]. Although the literature includes comprehensive studies examining various factors together (including breed) for pork [42], such studies are notably sparse for beef. The results of this study provide crucial data for the quantitative assessment of the effects of marinating beef.

### 3.3. WBSF

Tenderness is a crucial aspect of meat quality, significantly influencing consumers’ satisfaction and beef’s overall palatability. The proteolytic system, particularly calpains, has been highlighted as being significantly important for tenderness [43]. Numerous factors, including breed, slaughterhouse conditions, postmortem changes, and genetics, directly influence meat tenderness [1,17,22,44]. Given that tenderness, like beef color, significantly impacts consumers’ decisions, various initiatives have been undertaken to enhance this critical aspect of meat quality. Marination is one such initiative. The tenderizing effect of acidic marinades on meat is understood to involve multiple factors. These include the weakening of muscle fiber structures facilitated by the swelling of meat tissues. Additionally, the acidic environment enhances proteolysis, particularly through the activation of cathepsins, which are enzymes that break down proteins. Furthermore, acidic conditions promote the conversion of collagen, a tough protein in connective tissues, to gelatin during cooking. This transformation occurs more efficiently at lower pH levels, contributing to meat tenderness. These combined actions make acidic marinades effective in softening meat, improving its texture, and ease of consumption [2,9]. In this study, we systematically investigated the effects of four distinct marination treatments. The effect of marination time on WBSF values (both slope and total work) showed statistical significance in *m. longissimus thoracis et lumborum* (*p* < 0.001) and *m. semimembranosus* (*p* < 0.05), as shown in Table 1 and Table 2, respectively. In the longissimus muscle, the control samples exhibited the highest WBSF values. These values decreased, reaching the lowest level in samples marinated for 12 h (Table 3). Conversely, an increasing trend was observed in samples marinated for 24 h and 72 h (Figure 4). In *m. semimembranosus*, WBSF values increased, peaking at 12 h, before decreasing to match those of the control samples by 24 h (Figure 5). Furthermore, Charolais samples exhibited the highest mean WBSF slopes (5.22 ± 0.39 N/mm). The treatment type was significantly more effective in the semimembranosus muscle, as shown in Table 2. Samples from all marination methods exhibited lower WBSF values than the control samples (Table 4). Notably, the lowest WBSF values were associated with lemon marination, aligning with the anticipated effects of acidic marination practices. As indicated by Burke et al. [2], the decrease in WBSF values observed after marination with citric acid can be attributed to the impact of pH on water-binding capacity. Lowering the pH below the isoelectric point of muscle proteins causes protonation of the negatively charged –COO groups on protein molecules and disruption of some electrostatic bonds with –NH^3+^ groups on adjacent protein chains. This increase in the net positive charge likely leads to repulsion among protein groups with similar charges, thereby creating space for the immobilization of additional water [45]. Similar observations were evident for acetic acid as the tenderizing agent [2,46]. Similar to the observations in the longissimus muscle, the toughest beef was from the Charolais breed for *m. semimembranosus*; however, this difference was not statistically significant (Table 4). As shown in Appendix A, Aberdeen Angus longissimus samples marinated for 12 h were found to be the most tender (*p* < 0.01). The other two-way interactions were not significant for this muscle. Interestingly, none of the interactions were found to be statistically significant for WBSF in the semimembranosus muscle (Appendix A). Concerning the panel assessment, the highest scores for tenderness were given to Aberdeen Angus beef treated with olive oil−garlic marinade (Appendix A), lemon marination for 12 h, and olive oil−garlic marination for 24 h (Appendix A) in *m. longissimus thoracis et lumborum* (*p* < 0.05). In the semimembranosus muscle, comparable results were observed regarding beef tenderness. Aberdeen Angus subjected to olive oil-garlic marination (Appendix A) and the 24 h lemon and olive oil-garlic marination treatments (Appendix A) had the highest scores from panelists (*p* < 0.05). Soft meat is undoubtedly a characteristic favored by consumers. Marination, particularly with acidic solutions, plays a significant role in developing this attribute [2]. This process involves numerous biochemical reactions.

The collagen solubility data suggest that in addition to the dilution of the load due to swelling, solubilization of collagen may also contribute to tenderization following marination, especially in muscles with a high content of connective tissue [2]. The exact process by which collagen in fibers becomes soluble is not fully understood, but it is hypothesized to involve either hydrolysis of peptide bonds or slow breaking of covalent cross-links. The hydrolysis of peptide bonds accelerates significantly when the pH decreases from 6 to 4, which might explain the effectiveness of acid marinades in tenderizing meat. Alternatively, collagen may become solubilized through the degradation of cross-links, including unstable aldimine bonds (Schiff bases) affected by pH, heat, or denaturing agents, and other more stable links that may also break down under acidic conditions [47]. Moreover, reducing the pH of meat through acid marinades may enhance the proteolytic activity of cathepsins, potentially leading to more tender beef [2]. As summarized above, this biochemical process is highly dynamic and directly influenced by pH levels. This study elucidates the biological reasons why lemon and olive oil-garlic marinades are generally associated with more favorable parameters.

### 3.4. Panel Assessment

The fundamental objective of these applications to enhance beef quality is to cater to the tastes and preferences of consumers, ultimately leading to more appealing products. Therefore, panel evaluations are invaluable in practice, as they directly mirror consumer preferences and provide crucial data. In our study, the olive oil-garlic marinade was consistently the most favored across all traits for both types of muscle (Table 5 and Table 6). In *m. semimembranosus*, 24 h marination received the highest scores from the panelists for all traits, as shown in Table 6. Additionally, a notable finding was that Aberdeen Angus beef was consistently the most preferred according to panel assessments (Table 5 and Table 6). Olive oil−garlic marination of Aberdeen Angus beef had the significantly highest scores for color in the longissimus muscle (Appendix A). The marination treatment × marinating time interaction significantly influenced the panel evaluations, as detailed in Appendix A. Nonetheless, the olive oil-garlic marinade was the most favorable for the majority of traits assessed by the sensory panel (Appendix A). Furthermore, the results of the three-way interaction in our study also confirmed this finding. Aberdeen Angus beef marinated for 24 h in olive oil-garlic had the highest mean panel score (7.14 ± 0.40). Our study demonstrated that marination, particularly with olive oil, garlic, and lemon, is effective for the semimembranosus muscle, which is considered less valuable than the *m. longissimus thoracis et lumborum*. Burke et al. [2] suggested that citrus juice marination can effectively tenderize beef, including beef with a high content of connective tissue. These researchers also noted that since beef that is high in connective tissue generally has a lower economic value, their results suggest that marination may provide a way to enhance the value of these less expensive cuts. Our findings corroborated this suggestion and further demonstrated that the olive oil−garlic marination yielded even better outcomes, particularly in Aberdeen Angus meat.

Marinating meat with garlic and onion is a traditional preparation method before grilling over burning charcoal or wood in Mediterranean countries. Garlic contains many compounds with sulfhydryl groups, which are primarily formed through enzymatic reactions after mincing. In addition, the sugar content of garlic can influence its color as a consequence of the Maillard reaction [10]. This marination process may also inhibit the formation of heterocyclic aromatic amines, which are considered possible or probable carcinogens. It is recommended that humans minimize their exposure to these compounds [10,48]. Indeed, olive oil and garlic are key components of the Mediterranean diet, which is widely recognized as part of one of the healthiest lifestyles. Taken together, it can be asserted that olive oil-garlic marination not only enhances beef quality and sensory attributes, but also may have positive health advantages.

Although our study yielded significant methodological and practical recommendations, it had several notable limitations. Firstly, while three time points—12 h, 24 h, and 72 h—were examined in the longissimus muscle, only two—12 h and 24 h—were assessable in the semimembranosus muscle. A larger sample size may have facilitated a more thorough evaluation. Secondly, the marinades used, particularly olive oil and garlic, possess recognized antimicrobial properties [2,3,14,15,16,27]. This suggests that the parameters of meat quality might have been effectively monitored over an extended period using microbiological analyses.

## 4. Conclusions

Recently, the study of beef marination has gained significant attention in food science research. It is crucial to explore the roles of various marinade components and how they interact to effectively optimize the marination process. This deeper understanding will help tailor marinade formulations to enhance the desired qualities of beef products, such as flavor, tenderness, and moisture retention, thereby improving consumer satisfaction and product value. This study provides valuable data on the complex dynamics of beef marination across diverse scenarios and settings. These findings suggest that the cattle breed plays a crucial role in determining the success of beef marination techniques. Beef from various breeds exhibited distinct responses to different marination treatments, underscoring the critical importance of considering breed factors in beef marination for the first time. Additionally, sensory evaluation revealed that the olive oil-garlic marinade was favored by the panel for both muscle types examined. While many studies have primarily concentrated on the effects of marination, our approach encompasses a broader analysis, accounting for all potential factors influencing meat quality throughout the process from the slaughterhouse to the plate. Further molecular analysis could explore the biochemical mechanisms driving beef quality changes during marination and their subsequent effects on meat quality attributes. Specifically, the genetic background of cattle significantly affects beef quality parameters. Thus, exploring the interactions between an animal’s genetic structure and its response to marination may yield novel and intriguing findings.

## Figures and Tables

**Figure 1 foods-13-02979-f001:**
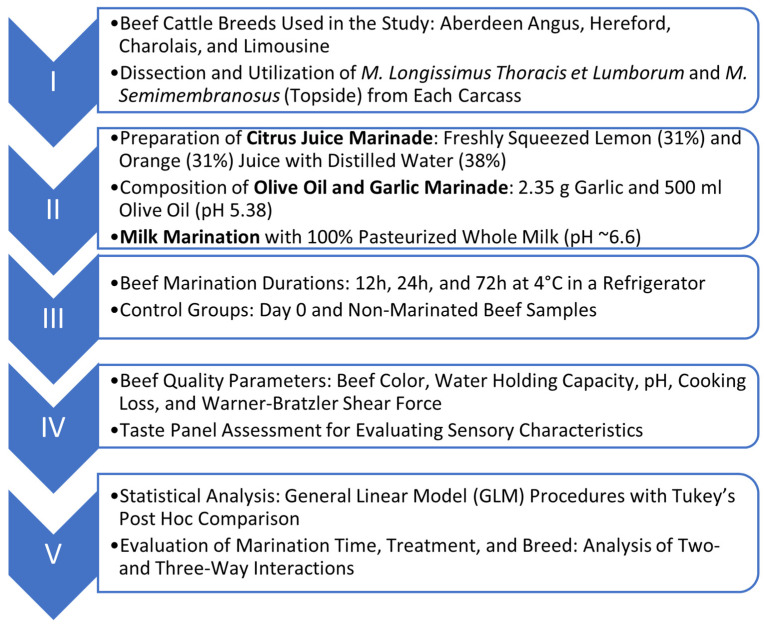
Schematic diagram of the research. The research work packages are indicated using Roman numerals.

**Figure 2 foods-13-02979-f002:**
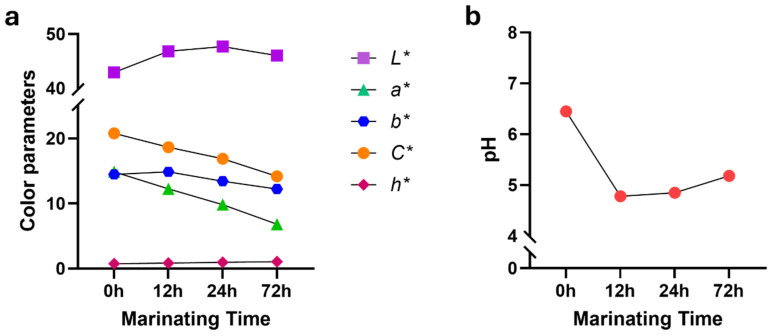
Relationships between marinating time, beef color, and pH in the *m. longissimus thoracis et lumborum* are explored. The changes in color parameters and pH dependent on time are depicted. (**a**) The alterations in color parameters, including *L** (lightness), *a** (redness), *b** (yellowness), *C** (chroma), and *h** (hue angle) based on marination time. (**b**) pH changes according to different marination times.

**Figure 3 foods-13-02979-f003:**
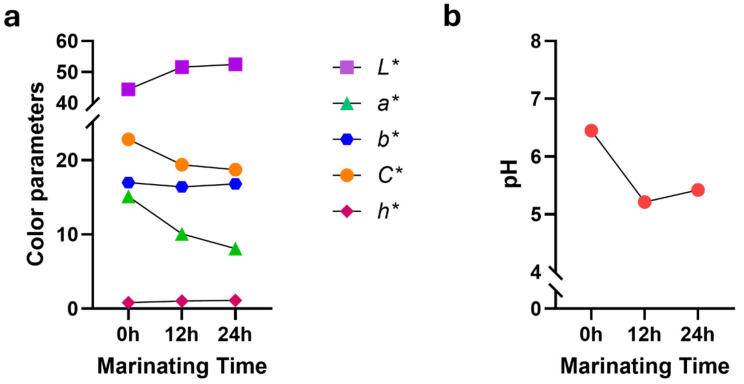
Effects of marination time on beef color and pH levels in the *m. semimembranosus* (topside). The figure illustrates how color parameters and pH levels change over time. (**a**) Variations in color parameters such as *L** (lightness), *a** (redness), *b** (yellowness), *C** (chroma), and *h** (hue angle) are analyzed in relation to marinating time. (**b**) Changes in pH are presented across varying marinating durations.

**Figure 4 foods-13-02979-f004:**
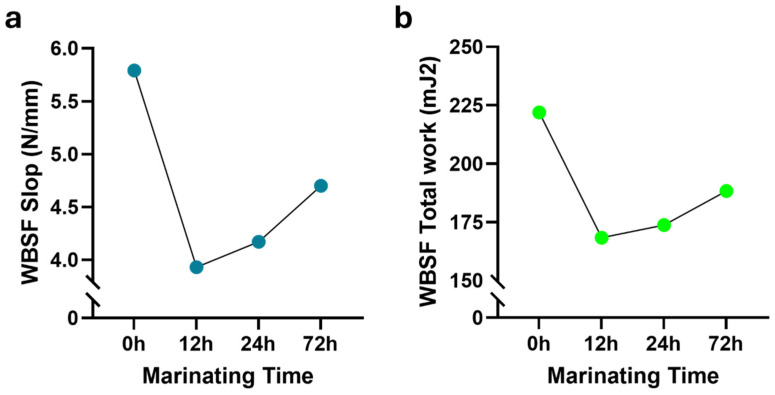
Impact of marination duration on Warner–Bratzler Shear Force (WBSF) measurements in the *m. longissimus thoracis et lumborum*. (**a**) WBSF slope (N/mm). (**b**) WBSF total work (mJ^2^). Shearing was performed using a Universal Testing Machine (Zwick/Roell Z0.5, Ulm, Germany) equipped with a V-shaped Warner–Bratzler attachment featuring a 60° triangular aperture.

**Figure 5 foods-13-02979-f005:**
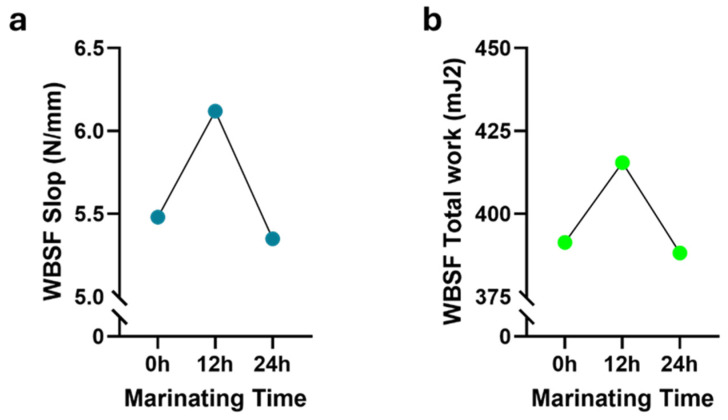
Effects of marinating time on Warner–Bratzler Shear Force (WBSF) levels in the *m. semimembranosus* (Topside). (**a**) WBSF slope (N/mm). (**b**) WBSF total work (mJ^2^). Shearing was performed using a Universal Testing Machine (Zwick/Roell Z0.5, Ulm, Germany), equipped with a V-shaped Warner–Bratzler attachment featuring a 60° triangular aperture.

**Table 1 foods-13-02979-t001:** Significant probability values obtained in the ANOVA for *m. longissimus thoracis et lumborum*.

Effect	Meat Color Parameters								WBSF Values ^1^			Taste Panel Assessment						
	*L**	*a**	*b**	*C**	*h*°	WHC (%)	pH	CL (%)	Peak Force (N)	Slope (N/mm)	Total Work (mJ^2^)	Odor	Flavor	Tenderness	Juiciness	Color	General Acceptance	Overall Liking
Time	<0.001	<0.001	<0.001	<0.001	<0.001	<0.001	<0.001	<0.001	<0.01	<0.001	<0.001	NS	<0.01	NS	NS	<0.05	NS	<0.05
Breed	NS	NS	NS	NS	NS	NS	NS	NS	<0.001	<0.05	NS	NS	<0.01	<0.001	<0.001	NS	<0.05	<0.05
Treatment	<0.05	<0.01	NS	<0.05	<0.001	NS	<0.001	<0.05	NS	NS	NS	<0.001	<0.001	<0.001	<0.001	<0.001	<0.001	<0.001
Time × Treatment	<0.001	<0.05	<0.01	<0.05	<0.01	NS	<0.001	<0.05	NS	NS	NS	NS	<0.05	NS	NS	NS	NS	NS
Time × Breed	NS	NS	NS	NS	NS	NS	NS	NS	NS	<0.01	NS	NS	NS	NS	NS	NS	NS	<0.05
Breed × Treatment	NS	NS	NS	NS	NS	NS	NS	NS	NS	NS	NS	NS	NS	NS	NS	NS	NS	NS
Time × Treatment × Breed	NS	NS	NS	NS	NS	NS	<0.05	NS	NS	<0.05	<0.01	NS	NS	NS	NS	NS	NS	<0.05

*L**: lightness, *a**: redness, *b**: yellowness, *C**: chroma, *h*°: hue angle. WHC: water holding capacity, CL: cooking loss, WBSF: Warner–Bratzler shear force. ^1^ Peak force: force needed to shear through the meat; slope: slope between the origin and peak of the curve; total work: total energy needed to shear through the meat. NS: not significant.

**Table 2 foods-13-02979-t002:** Significant probability values obtained in the ANOVA for *m. semimembranosus* (topside).

Effect	Meat Color Parameters								WBSF Values ^1^			Taste Panel Assessment						
	*L**	*a**	*b**	*C**	*h*°	WHC (%)	pH	CL (%)	Peak Force (N)	Slope (N/mm)	Total Work (mJ^2^)	Odor	Flavor	Tenderness	Juiciness	Color	General Acceptance	Overall Liking
Time	<0.001	<0.001	<0.05 ǂ	<0.001	<0.001	<0.001	<0.001	<0.01	<0.01	<0.05	<0.05	<0.05	<0.001	<0.001	<0.01	<0.001	<0.001	<0.001
Breed	NS	NS	<0.05	<0.05	NS	<0.05 ǂ	NS	<0.01	<0.001	NS	<0.05 ǂ	NS	<0.05	<0.001	<0.001	<0.05	NS	<0.01
Treatment	<0.05 ǂ	NS	NS	NS	NS	<0.001	<0.01	<0.05 ǂ	NS	<0.01	<0.001	<0.001	<0.001	<0.001	<0.001	<0.001	<0.001	<0.001
Time × Treatment	<0.05	NS	NS	NS	NS	<0.01	<0.01	NS	NS	NS	NS	<0.01	<0.01	<0.05	NS	<0.001	<0.05	<0.01
Time × Breed	NS	<0.05	<0.05	<0.05	NS	NS	<0.05	NS	NS	NS	NS	NS	NS	NS	NS	NS	NS	NS
Breed × Treatment	NS	NS	NS	NS	NS	NS	NS	NS	NS	NS	NS	NS	NS	<0.05	NS	<0.05	NS	NS
Time × Treatment × Breed	NS	NS	NS	NS	NS	<0.05	NS	NS	NS	NS	NS	NS	NS	NS	NS	NS	NS	NS

*L**: lightness, *a**: redness, *b**: yellowness, *C**: chroma, *h*°: hue angle. ǂ Not confirmed by the Tukey multiple comparison. WHC: water holding capacity, CL: cooking loss, WBSF: Warner–Bratzler shear force. ^1^ Peak force: force needed to shear through the meat; slope: shear firmness or the slope between the origin and the peak of the curve; total work: total energy needed to shear through the meat. NS: not significant.

**Table 3 foods-13-02979-t003:** Least-squares means and standard errors for marination time, marination treatment, and breed on beef quality parameters in *m. longissimus thoracis et lumborum*.

Trait		Marinating Time			Marination Treatment				Breed			
	0 h Control	12 h	24 h	72 h	Non-Marinate	Lemon	Milk	Olive Oil-Garlic	Hereford	Charolais	Aberdeen Angus	Limousine
***L****	**42.96 ± 0.49 ^b^**	**46.85 ± 0.69 ^a^**	**47.72 ± 0.88 ^a^**	**46.04 ± 0.89 ^a^**	**43.88 ± 1.20 ^b^**	**45.63 ± 1.20 ^ab^**	**48.60 ± 1.25 ^a^**	**45.45 ± 1.38 ^ab^**	47.54 ± 1.20	45.78 ± 1.25	45.57 ± 1.17	44.68 ± 1.41
***a****	**14.88 ± 0.37 ^a^**	**12.25 ± 0.65 ^a^**	**9.83 ± 0.64 ^ab^**	**6.84 ± 0.42 ^b^**	**13.36 ± 0.67 ^a^**	**10.66 ± 0.67 ^ab^**	**9.69 ± 0.69 ^b^**	**10.01 ± 0.77 ^b^**	10.69 ± 0.67	9.77 ± 0.69	11.56 ± 0.65	11.77 ± 0.79
***b****	**14.48 ± 0.37 ^a^**	**14.88 ± 0.35 ^a^**	**13.43 ± 0.29 ^ab^**	**12.23 ± 0.31 ^b^**	13.85 ± 0.39	13.56 ± 0.39	13.18 ± 0.41	13.43 ± 0.45	13.74 ± 0.39	12.76 ± 0.41	13.68 ± 0.38	13.86 ± 0.46
***C****	**20.79 ± 0.51 ^a^**	**18.65 ± 0.65 ^a^**	**16.89 ± 0.57 ^ab^**	**14.19 ± 0.42 ^b^**	**19.36 ± 0.65 ^a^**	**17.47 ± 0.65 ^ab^**	**16.62 ± 0.67 ^b^**	**17.08 ± 0.74 ^b^**	17.66 ± 0.65	16.28 ± 0.67	18.01 ± 0.63	18.48 ± 0.76
***h****	**0.77 ± 0.01 ^b^**	**0.87 ± 0.02 ^ab^**	**0.98 ± 0.03 ^a^**	**1.07 ± 0.02 ^a^**	**0.82 ± 0.03 ^b^**	**0.94 ± 0.03 ^a^**	**0.97 ± 0.02 ^a^**	**0.96 ± 0.03 ^a^**	0.94 ± 0.03	0.95 ± 0.03	0.90 ± 0.02	0.90 ± 0.03
**WHC (%)**	**17.45 ± 0.42 ^a^**	**16.86 ± 0.30 ^a^**	**16.16 ± 0.64 ^a^**	**9.80 ± 0.39 ^b^**	14.40 ± 0.55	15.76 ± 0.55	14.91 ± 0.57	15.21 ± 0.63	14.50 ± 0.55	14.36 ± 0.57	16.02 ± 0.53	15.39 ± 0.64
**pH ^1^**	**6.45 ± 0.03 ^a^**	**4.78 ± 0.05 ^b^**	**4.85 ± 0.07 ^b^**	**5.18 ± 0.11 ^ab^**	**5.49 ± 0.10 ^a^**	**4.90 ± 0.10 ^b^**	**5.58 ± 0.11 ^a^**	**5.28 ± 0.12 ^a^**	5.34 ± 0.10	5.31 ± 0.11	5.19 ± 0.10	5.42 ± 0.12
**CL**	**31.38 ± 0.75 ^b^**	**34.89 ± 0.87 ^ab^**	**35.87 ± 0.79 ^a^**	**33.44 ± 0.98 ^ab^**	**30.94 ± 1.09 ^b^**	**35.89 ± 1.09 ^a^**	**35.14 ± 1.14 ^ab^**	**33.49 ± 1.25 ^ab^**	33.56 ± 1.09	35.17 ± 1.14	34.90 ± 1.06	31.96 ± 1.28
**WBSF Peak Force (N)**	**64.32 ± 3.24 ^a^**	**30.36 ± 1.66 ^b^**	**53.27 ± 2.95 ^ab^**	**38.19 ± 4.34 ^ab^**	46.03 ± 2.62	42.34 ± 4.01	51.17 ± 4.60	42.27 ± 4.77	**34.46 ± 2.74 ^b^**	**51.84 ± 2.64 ^a^**	**39.67 ± 2.44 ^b^**	**36.14 ± 2.91 ^b^**
**WBSF Slope (N/mm)**	**5.79 ± 0.13 ^a^**	**3.93 ± 0.34 ^b^**	**4.17 ± 0.40 ^b^**	**4.70 ± 0.26 ^ab^**	4.43 ± 0.37	4.77 ± 0.36	5.26 ± 0.39	4.13 ± 0.43	**4.71 ± 0.37 ^b^**	**5.22 ± 0.39 ^a^**	**4.29 ± 0.36 ^b^**	**4.18 ± 0.44 ^b^**
**WBSF Total work (mJ^2^)**	221.89 ± 10.95	168.28 ± 17.52	173.73 ± 15.56	188.33 ± 18.30	200.65 ± 23.03	174.41 ± 23.02	191.16 ± 23.96	189.02 ± 26.43	213.91 ± 23.03	204.91 ± 23.96	193.33 ± 22.34	180.09 ± 27.02

^1^ pH measurement for the 0 h control was performed at 1 h postmortem. *L**: lightness, *a**: redness, *b**: yellowness, *C**: chroma, *h**: hue angle. WHC: water holding capacity; CL: cooking loss; WBSF: Warner–Bratzler shear force. ^a,b^ Different superscripts within a row for each factor indicate significant difference.

**Table 4 foods-13-02979-t004:** Least-squares means and standard errors for marinating time, marination treatment, and breed on beef quality parameters in *m. semimembranosus* (topside).

Trait		Marinating Time		Marination Treatment				Breed			
	0 h Control	12 h	24 h	Non-Marinate	Lemon	Milk	Olive Oil-Garlic	Hereford	Charolais	Aberdeen Angus	Limousine
***L****	**44.39 ± 0.78 ^b^**	**51.57 ± 0.63 ^a^**	**52.44 ± 0.68 ^a^**	48.71 ± 1.00	47.64 ± 1.01	51.77 ± 1.04	49.75 ± 1.15	50.81 ± 1.00	50.14 ± 1.04	47.78 ± 0.97	49.14 ± 46.75
***a****	**15.11 ± 0.38 ^a^**	**10.07 ± 0.51 ^b^**	**8.09 ± 0.25 ^b^**	11.31 ± 0.43	10.71 ± 0.43	11.37 ± 0.45	10.96 ± 0.49	11.93 ± 0.43	10.79 ± 0.45	10.77 ± 0.42	10.86 ± 0.50
***b****	16.96 ± 0.59	16.41 ± 0.34	16.80 ± 0.23	16.71 ± 0.56	16.09 ± 0.56	16.63 ± 0.58	17.47 ± 0.64	**18.15 ± 0.56 ^a^**	**16.26 ± 0.58 ^ab^**	**15.56 ± 0.54 ^b^**	**15.99 ± 0.66 ^b^**
***C****	**22.79 ± 0.65 ^a^**	**19.37 ± 0.53 ^b^**	**18.72 ± 0.21 ^b^**	20.36 ± 0.62	19.54 ± 0.62	20.38 ± 0.65	20.89 ± 0.71	**21.96 ±** **0.62 ^a^**	**19.87 ± 0.65 ^ab^**	**19.09 ± 0.60 ^b^**	**19.69 ± 0.73 ^ab^**
***h****	**0.83 ± 0.01 ^b^**	**1.03 ± 0.02 ^ab^**	**1.12 ± 0.01 ^a^**	0.98 ± 0.02	0.99 ± 0.01	0.98 ± 0.02	1.01 ± 0.02	0.99 ± 0.02	1.00 ± 0.02	0.97 ± 0.01	1.00 ± 0.02
**WHC (%)**	**15.91 ± 0.38 ^a^**	**9.50 ± 0.32 ^b^**	**10.54 ± 0.65 ^ab^**	**11.34 ± 0.50 ^b^**	**11.14 ± 0.50 ^b^**	**13.96 ± 0.52 ^a^**	**11.50 ± 0.57 ^b^**	12.06 ± 0.49	13.16 ± 0.52	11.92 ± 0.48	10.80 ± 0.58
**pH ^1^**	**6.45 ± 0.03 ^a^**	**5.21 ± 0.07 ^b^**	**5.42 ± 0.09 ^ab^**	**5.85 ± 0.09 ^a^**	**5.37 ± 0.09 ^b^**	**5.82 ± 0.09 ^a^**	**5.72 ± 0.11 ^ab^**	5.76 ± 0.09	5.72 ± 0.10	5.46 ± 0.09	5.82 ± 0.11
**CL (%)**	**36.81 ± 0.20 ^b^**	**38.40 ± 0.29 ^a^**	**35.83 ± 0.68 ^b^**	35.68 ± 0.51	36.27 ± 0.51	37.96 ± 0.53	38.13 ± 0.59	**38.14 ± 0.51 ^a^**	**37.51 ± 0.53 ^a^**	**35.40 ± 0.50 ^b^**	**36.99 ± 0.60 ^ab^**
**WBSF Peak Force (N)**	**67.32 ± 1.21 ^a^**	**54.23 ± 1.51 ^b^**	**63.76 ± 1.70 ^a^**	54.02 ± 2.74	52.40 ± 3.21	58.09 ± 2.29	59.50 ± 2.70	**51.55 ± 1.99 ^a^**	**55.21 ± 2.22 ^a^**	**47.45 ± 2.19 ^b^**	**47.44 ± 1.93 ^b^**
**WBSF Slope (N/mm)**	**5.48 ± 0.25 ^b^**	**6.12 ± 0.22 ^a^**	**5.35 ± 0.19 ^b^**	**6.72 ± 0.30 ^a^**	**5.17 ± 0.31 ^b^**	**5.85 ± 0.25 ^ab^**	**5.40 ± 0.27 ^b^**	5.52 ± 0.25	5.93 ± 0.28	5.71 ± 0.29	5.39 ± 0.33
**WBSF Total work (mJ^2^)**	**391.44 ± 15.80 ^b^**	**415.48 ± 22.02 ^a^**	**388.22 ± 18.72 ^b^**	**450.52 ± 20.51 ^a^**	**252.81 ± 17.77 ^b^**	**347.15 ± 18.72 ^ab^**	**309.83 ± 19.05 ^b^**	346.86 ±16.27	339.81 ± 17.79	344.52 ± 18.71	340.51 ± 19.18

^1^ pH measurement for 0 h control was performed at 1 h postmortem. *L**: lightness, *a**: redness, *b**: yellowness, *C**: chroma, *h**: hue angle. WHC: water holding capacity; CL: cooking loss; WBSF: Warner–Bratzler shear force. ^a,b^ Different superscripts within a row for each factor indicate significant difference.

**Table 5 foods-13-02979-t005:** Least-squares means and standard errors for marinating time, marination treatment, and breed on sensory traits in *m. longissimus thoracis et lumborum*.

Trait	Marinating Time		Marination Treatment				Breed			
	12 h	24 h	Non-Marinate	Lemon	Milk	Olive Oil-Garlic	Hereford	Charolais	Aberdeen Angus	Limousine
**Odor**	4.99 ± 0.09	4.93 ± 0.08	**4.97 ± 0.13 ^ab^**	**4.70 ± 0.12 ^b^**	**4.79 ± 0.15 ^b^**	**5.38 ± 0.13 ^a^**	4.98 ± 0.13	4.77 ± 0.18	5.18 ± 0.12	4.91 ± 0.13
**Flavor**	**5.25 ± 0.08 ^a^**	**4.87 ± 0.09 ^b^**	**4.82 ± 0.13 ^b^**	**4.82 ± 0.13 ^b^**	**4.86 ± 0.13 ^b^**	**5.75 ± 0.12 ^a^**	**5.14 ± 0.13 ^ab^**	**4.72 ± 0.12 ^b^**	**5.33 ± 0.12 ^a^**	**5.06 ± 0.13 ^ab^**
**Tenderness**	4.91 ± 0.03	4.89 ± 0.09	**4.26 ± 0.13 ^b^**	**5.29 ± 0.11 ^a^**	**4.67 ± 0.11 ^b^**	**5.39 ± 0.17 ^a^**	**5.04 ± 0.13 ^a^**	**4.38 ± 0.13 ^b^**	**5.24 ± 0.21 ^a^**	**4.94 ± 0.14 ^a^**
**Juiciness**	4.64 ± 0.09	4.63 ± 0.09	**4.09 ± 0.12 ^b^**	**5.04 ± 0.13 ^a^**	**4.27 ± 0.11 ^b^**	**5.14 ± 0.13 ^a^**	**4.94 ± 0.18 ^a^**	**4.05 ± 0.13 ^b^**	**4.92 ± 0.11 ^a^**	**4.63 ± 0.09 ^a^**
**Color**	**5.19 ± 0.09 ^b^**	**5.50 ± 0.10 ^a^**	**5.38 ± 0.13 ^b^**	**5.19 ± 0.13 ^bc^**	**4.79 ± 0.16 ^c^**	**6.03 ± 0.12 ^a^**	5.42 ± 0.13	5.10 ± 0.16	5.56 ± 0.12	5.31 ± 0.12
**General Acceptance**	4.98 ± 0.05	4.80 ± 0.09	**4.76 ± 0.13 ^bc^**	**4.42 ± 0.17 ^c^**	**4.95 ± 0.13 ^b^**	**5.45 ± 0.14 ^a^**	**4.94 ± 0.13 ^ab^**	**4.60 ± 0.14 ^b^**	**5.16 ± 0.14 ^a^**	**4.86 ± 0.13 ^ab^**
**Overall liking**	**5.05 ± 0.09 ^a^**	**4.75 ± 0.09 ^b^**	**4.83 ± 0.14 ^b^**	**4.44 ± 0.13 ^b^**	**4.86 ± 0.14 ^b^**	**5.47 ± 0.13 ^a^**	**4.87 ± 0.13 ^ab^**	**4.62 ± 0.11 ^b^**	**5.16 ± 0.15 ^a^**	**4.96 ± 0.13 ^ab^**

The assessment was conducted using an 8-point hedonic scale. General acceptance (8 = extremely desirable, 1 = extremely undesirable, abnormal flavor or odor), and overall liking (8 = like extremely, very definitely would purchase, 1 = dislike extremely, very definitely would not purchase). ^a,b,c^ Different superscripts within a row for each factor indicate significant difference.

**Table 6 foods-13-02979-t006:** Least-squares means and standard errors for marinating time, marination treatment, and breed on sensory traits in *m. semimembranosus* (topside).

Trait	Marinating Time		Marination Treatment				Breed			
	12 h	24 h	Non-Marinate	Lemon	Milk	Olive Oil-Garlic	Hereford	Charolais	Aberdeen Angus	Limousine
**Odor**	**4.37 ± 0.10 ^b^**	**4.74 ± 0.11 ^a^**	**4.23 ± 0.16 ^b^**	**4.43 ± 0.16 ^b^**	**4.34 ± 0.15 ^b^**	**5.22 ± 0.17 ^a^**	4.49 ± 0.15	4.33 ± 0.16	4.78 ± 0.16	4.61 ± 0.15
**Flavor**	**4.39 ± 0.10 ^b^**	**4.98 ± 0.13 ^a^**	**4.16 ± 0.15 ^b^**	**5.01 ± 0.15 ^a^**	**4.28 ± 0.17 ^b^**	**5.29 ± 0.14 ^a^**	**4.42 ± 0.14 ^b^**	**4.58 ± 0.15 ^ab^**	**4.95 ± 0.15 ^a^**	**4.78 ± 0.14 ^ab^**
**Tenderness**	**4.39 ± 0.11 ^b^**	**4.88 ± 0.12 ^a^**	**4.22 ± 0.15 ^b^**	**5.16 ± 0.13 ^a^**	**4.23 ± 0.14 ^b^**	**4.95 ± 0.15 ^a^**	**4.09 ± 0.15 ^c^**	**4.46 ± 0.16 ^b^**	**5.10 ± 0.16 ^a^**	**4.90 ± 0.15 ^ab^**
**Juiciness**	**4.19 ± 0.11 ^b^**	**4.62 ± 0.10 ^a^**	**3.91 ± 0.15 ^d^**	**4.88 ± 0.19 ^a^**	**4.14 ± 0.15 ^c^**	**4.69 ± 0.11 ^b^**	**3.85 ± 0.15 ^d^**	**4.30 ± 0.15 ^c^**	**4.85 ± 0.17 ^a^**	**4.61 ± 0.11 ^b^**
**Color**	**4.44 ± 0.10 ^b^**	**5.02 ± 0.11 ^a^**	**4.51 ± 0.14 ^c^**	**4.95 ± 0.14 ^b^**	**4.21 ± 0.11 ^d^**	**5.25 ± 0.14 ^a^**	**4.39 ± 0.14 ^b^**	**4.66 ± 0.15 ^ab^**	**4.94 ± 0.13 ^a^**	**4.92 ± 0.14 ^a^**
**General Acceptance**	**4.25 ± 0.14 ^b^**	**4.91 ± 0.10 ^a^**	**4.28 ± 0.15 ^b^**	**4.28 ± 0.15 ^b^**	**4.43 ± 0.14 ^b^**	**5.32 ± 0.15 ^a^**	4.39 ± 0.14	4.48 ± 0.11	4.85 ± 0.15	4.59 ± 0.15
**Overall liking**	**4.33 ± 0.11 ^b^**	**4.94 ± 0.10 ^a^**	**4.36 ± 0.15 ^b^**	**4.34 ± 0.17 ^b^**	**4.42 ± 0.15 ^b^**	**5.42 ± 0.14 ^a^**	**4.28 ± 0.14 ^b^**	**4.54 ± 0.15 ^ab^**	**4.98 ± 0.15 ^a^**	**4.74 ± 0.14 ^ab^**

The assessment was conducted using an 8-point hedonic scale. General acceptance (8 = extremely desirable, 1 = extremely undesirable, abnormal flavor or odor), overall liking (8 = like extremely, very definitely would purchase, 1 = dislike extremely, very definitely would not purchase). ^a,b,c,d^ Different superscripts within a row for each factor indicate significant difference.

**Table 7 foods-13-02979-t007:** Least-squares means and standard errors for the effect of breed × marinating time × marination treatment interaction on beef pH in *m. longissimus thoracis et lumborum* (*p* < 0.05).

Breed	Marinating Time	Marination Treatment			
		Non-Marinate Control	Lemon	Milk	Olive Oil-Garlic
**Hereford**	0 h	6.68 ± 0.08			
	12 h	4.98 ± 0.16	4.56 ± 0.19	5.12 ± 0.23	4.76 ± 0.19
	24 h	5.25 ± 0.22	4.52 ± 0.26	5.17 ± 0.32	4.54 ± 0.26
	72 h	5.69 ± 0.34	4.51 ± 0.39	5.64 ± 0.48	4.62 ± 0.29
**Charolais**	0 h	6.60 ± 0.09			
	12 h	4.65 ± 0.19	4.23 ± 0.23	5.11 ± 0.23	5.01 ± 0.15
	24 h	4.98 ± 0.26	3.95 ± 0.32	5.17 ± 0.32	5.13 ± 0.20
	72 h	5.10 ± 0.39	3.98 ± 0.48	5.64 ± 0.28	5.74 ± 0.30
**Aberdeen Angus**	0 h	6.26 ± 0.09			
	12 h	4.69 ± 0.19	4.14 ± 0.18	5.03 ± 0.19	4.61 ± 0.11
	24 h	4.90 ± 0.26	4.13 ± 0.25	5.22 ± 0.26	4.74 ± 0.26
	72 h	5.54 ± 0.39	4.24 ± 0.39	5.64 ± 0.39	4.83 ± 0.39
**Limousine**	0 h	6.35 ± 0.11			
	12 h	5.00 ± 0.23	4.69 ± 0.16	4.98 ± 0.15	4.98 ± 0.33
	24 h	5.26 ± 0.32	4.72 ± 0.22	5.19 ± 0.19	4.69 ± 0.45
	72 h	6.09 ± 0.48	5.11 ± 0.34	5.41 ± 0.30	5.06 ± 0.48

**Table 8 foods-13-02979-t008:** Least-squares means and standard errors for the effect of breed × marinating time × marination treatment interaction on the Warner–Bratzler shear force (WBSF) slope (N/mm) in *m. longissimus thoracis et lumborum* (*p* < 0.05).

Breed	Marinating Time	Marination Treatment			
		Non-Marinate Control	Lemon	Milk	Olive Oil-Garlic
**Hereford**	0 h	5.73 ± 0.42			
	12 h	3.23 ± 1.09	5.86 ± 1.26	6.28 ± 1.54	2.55 ± 1.26
	24 h	2.59 ± 1.28	4.76 ± 1.48	4.51 ± 1.81	2.23 ± 1.48
	72 h	5.17 ± 0.85	4.79 ± 0.98	8.73 ± 1.19	3.73 ± 0.98
**Charolais**	0 h	6.75 ± 0.48			
	12 h	4.41 ± 1.26	6.79 ± 1.54	**9.26 ± 1.54**	3.41 ± 0.97
	24 h	5.32 ± 1.48	7.11 ± 1.81	2.93 ± 1.81	3.29 ± 1.15
	72 h	5.05 ± 0.98	2.82 ± 1.19	3.86 ± 1.19	3.95 ± 0.076
**Aberdeen Angus**	0 h	5.39 ± 0.48			
	12 h	2.34 ± 1.26	3.08 ± 1.26	2.72 ± 1.26	**1.68 ± 1.26**
	24 h	4.75 ± 1.48	3.93 ± 1.48	5.71 ± 1.48	5.21 ± 1.48
	72 h	5.26 ± 0.98	4.87 ± 0.98	3.92 ± 0.98	2.92 ± 0.98
**Limousine**	0 h	5.07 ± 0.59			
	12 h	2.14 ± 1.54	2.00 ± 1.09	3.59 ± 0.97	3.54 ± 2.18
	24 h	3.86 ± 1.81	2.91 ± 1.28	4.97 ± 1.15	2.61 ± 2.56
	72 h	3.87 ± 1.19	3.45 ± 0.85	3.44 ± 0.76	**9.34 ± 1.69**

**Table 9 foods-13-02979-t009:** Least-squares means and standard errors for the effect of breed × marinating time × marination treatment interaction on Warner–Bratzler shear force (WBSF) and total work (mJ^2^) in *m. longissimus thoracis et lumborum* (*p* < 0.01).

Breed	Marinating Time	Marination Treatment			
		Non-Marinate Control	Lemon	Milk	Olive Oil-Garlic
**Hereford**	0 h	232.03 ± 15.10			
	12 h	149.98 ± 26.15	273.74 ± 24.84	283.12 ± 19.41	205.72 ± 24.84
	24 h	147.63 ± 19.87	164.64 ± 17.59	146.22 ± 20.53	230.79 ± 17.59
	72 h	132.52 ± 18.65	276.59 ± 27.72	**361.79 ± 22.94**	150.11 ± 17.72
**Charolais**	0 h	308.77 ± 20.53			
	12 h	208.74 ± 24.84	122.36 ± 19.41	127.45 ± 29.41	199.01 ± 20.23
	24 h	305.96 ± 17.59	144.20 ± 20.53	115.68 ± 20.53	195.48 ± 14.61
	72 h	218.76 ± 27.72	**98.32 ± 10.04**	163.39 ± 22.94	315.23 ± 22.46
**Aberdeen Angus**	0 h	186.42 ± 20.53			
	12 h	136.95 ± 24.84	154.02 ± 24.84	144.58 ± 14.84	95.22 ± 24.84
	24 h	218.41 ± 17.59	118.96 ± 27.59	123.78 ± 17.59	184.16 ± 17.59
	72 h	176.78 ± 27.72	149.49 ± 17.72	118.44 ± 67.72	118.59 ± 17.72
**Limousine**	0 h	216.63 ± 19.64			
	12 h	128.77 ± 12.41	119.59 ± 16.15	179.41 ± 20.23	163.88 ± 22.31
	24 h	190.38 ± 20.53	148.80 ± 19.87	255.35 ± 24.61	89.35 ± 19.74
	72 h	251.77 ± 22.94	173.36 ± 18.65	178.77 ± 22.46	169.42 ± 17.29

**Table 10 foods-13-02979-t010:** Least-squares means and standard errors for the effect of breed × marinating time × marination treatment interaction on water holding capacity (WHC) in *m. semimembranosus* (topside) (*p* < 0.05).

Breed	Marinating Time	Marination Treatment			
		Non-Marinate Control	Lemon	Milk	Olive Oil-Garlic
**Hereford**	0 h	14.06 ± 1.20			
	12 h	10.24 ± 1.01	8.29 ± 1.17	10.89 ± 1.43	8.02 ± 1.17
	24 h	9.69 ± 2.10	9.15 ± 2.42	**18.99 ± 1.96**	10.34 ± 2.42
**Charolais**	0 h	19.02 ± 1.39			
	12 h	10.39 ± 1.17	9.97 ± 1.43	10.06 ± 1.43	11.19 ± 0.91
	24 h	8.51 ± 2.42	8.79 ± 2.96	18.66 ± 2.96	12.60 ± 1.87
**Aberdeen Angus**	0 h	16.69 ± 1.39			
	12 h	9.11 ± 1.17	9.35 ± 1.17	10.81 ± 1.17	9.27 ± 1.17
	24 h	7.40 ± 2.42	8.08 ± 2.42	10.83 ± 2.42	11.95 ± 2.42
**Limousine**	0 h	16.83 ± 1.70			
	12 h	7.31 ± 1.43	7.86 ± 1.01	10.29 ± 0.91	9.02 ± 2.02
	24 h	6.86 ± 2.96	8.72 ± 2.10	13.37 ± 1.87	**4.63 ± 1.19**

**Table 11 foods-13-02979-t011:** Least-squares means and standard errors for the effect of breed × marinating time × marination treatment interaction on overall liking in the panel assessment of *m. longissimus thoracis et lumborum* (*p* < 0.05).

Breed	Marinating Time	Marination Treatment			
		Non-Marinate Control	Lemon	Milk	Olive Oil-Garlic
**Hereford**	12 h	4.71 ± 0.41	4.38 ± 0.39	5.14 ± 0.38	6.59 ± 0.39
	24 h	5.29 ± 0.37	**3.86 ± 0.40**	4.71 ± 0.42	4.76 ± 0.37
**Charolais**	12 h	4.24 ± 0.42	4.29 ± 0.38	4.43 ± 0.39	5.48 ± 0.42
	24 h	4.71 ± 0.39	4.38 ± 0.43	4.48 ± 0.38	4.95 ± 0.38
**Aberdeen Angus**	12 h	4.86 ± 0.38	4.57 ± 0.37	5.38 ± 0.40	5.71 ± 0.42
	24 h	4.95 ± 0.41	4.68.0.40	5.05 ± 0.39	**7.14 ± 0.40**
**Limousine**	12 h	5.14 ± 0.37	5.38 ± 0.39	5.09 ± 0.43	5.86 ± 0.38
	24 h	4.76 ± 0.42	4.01 ± 0.38	4.57 ± 0.39	4.76 ± 0.41

Overall liking was assessed using a market-oriented approach, where panelists were evaluated based on which product they would prefer to choose again (8 = like extremely, very definitely would purchase, 1 = dislike extremely, very definitely would not purchase).

## Data Availability

The original contributions presented in the study are included in the article and Appendix A, further inquiries can be directed to the corresponding author.

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
