# Peer review of "Unraveling the Complexities of Beef Marination: Effect of Marinating Time, Marination Treatments, and Breed"

_foods, 2024, doi:10.3390/foods13182979_

Round 1

Reviewer 1 Report

Comments and Suggestions for Authors

The manuscript foods-3163644 entitled "Toward unraveling the complexities of beef marination: The influences of marinating time, marination treatments, and breed on beef quality parameters and sensory traits in m. longissimus thoracis et lumborum and m. semimembranosus (topside)"  

The manuscript is well-written. I have some suggestions for the improvement.

Major recommendations:

1)     Why the authors selected milk, olive oil and garlic, and lemon juice as marinade? Please include this information in the introduction. Also, what is already done on this topic on marination time, marination type etc. need to be included in the introduction.

2)     Please also explain the line 54. Effect of marination on color in details.

3)     Citations brackets need to be corrected throughout the manuscript. Please replace () with [].

4)     2.1 Animals were off-fed or not? Please specify.

5)     2.1 Total number of animals slaughtered? Please specify.

6)     2.1 please specify the sex of animals. All males or mixed?

7)     Line 85: What time post-mortem animals were deboned? Please include

8)     Line 97: Specifications of vacuum packaging material and equipment?

9)     Line 97: Specifications of refrigerator?

10)  Line 99: What do you mean by three times repeated measurements? Experiment was independently repeated three times? Or three recordings were taken?

11)   Please include a table indicating treatments.

12)  Line 114: "Moreover, the chroma value (C*) and hue angle (hº) were calculated according to the formulas as follows". Please give citation here.

13)  2.4 Water holding capacity: Please provide citation.

14)  2.8 Sensory analysis: How much sensory sessions were conducted? Or all samples were tested in a single session? In that case, how much total time was taken to conduct this study? It was a single sitting testing? Please specify.

15)  Sensory analysis: For general acceptance and overall liking, the panel should be large enough to reach to a conclusion. I will recommend to delete these.

16)   Why the interactions of muscle types were studied?

17)  Abbreviate WHC, WBSF, and CL when first time these appeared in the manuscript other than abstract. Please check it.

18)   Lines 14-15: Please italicize muscle names here and check these throughout the manuscript.

19)  Line 18: Please correct " cook loss" as "cooking loss"

20)  Line 20: Please rephrase sentence. Do not use "We". Rephrase

21)  Line 39: Correct "marinating" with "marination"

Line 94: Please write author name or first author et al when to mention such like according to, etc. Please correct this throughout the manuscript.

Author Response

General comment:

The manuscript is well-written. I have some suggestions for improvement.

Response:

We thank the reviewer. We considered all of the suggestions and adapted them to the paper.

Comment 1:

Why the authors selected milk, olive oil and garlic, and lemon juice as marinade? Please include this information in the introduction. Also, what is already done on this topic on marination time, marination type etc. need to be included in the introduction.

Response 1:

We thank the reviewer and agree with their suggestions. Therefore, we have expanded the paragraph that outlines the study's objectives in the final part of the introduction section (Lines 68-79). The included part is as follows:

“In choosing the marinade components for this study—milk, olive oil, garlic, and lemon juice—we considered both traditional culinary practices and their scientifically documented impacts on meat quality. Dairy products are used for its tenderizing effect due to the presence of calcium and lactic acid, which activates enzymes that break down proteins [12]. Olive oil is selected for its fat content that enhances flavor infusion and moisture retention in the meat [13]). Garlic and lemon juice are known for their strong antimicrobial properties and their ability to enhance meat flavor and tenderness through acidity which aids in the breakdown of meat fibers [2, 6,11]. Previous research combining these natural marinades has demonstrated enhancements in meat safety, including microbial quality and fat oxidation, as well as improvements in physical characteristics like color and texture, as determined through instrumental measurements [12]. However, there is no comprehensive study in the literature that addresses these marinations along with marination time and the breed factors of the beef used.”

Comment 2:

Please also explain the line 54. Effect of marination on color in details.

Response 2:

We thank the reviewer for the clarification. A new part indicating the marination on color is now added (58-67).

Comment 3:

Citations brackets need to be corrected throughout the manuscript. Please replace () with [].

Response 3:

We thank the reviewer for the correction. In-text citations are now revised throughout the manuscript.

Comment 4:

2.1 Animals were off-fed or not? Please specify.

Response 4:

We thank the reviewer for the clarification. We now added the corresponding information (Lines 107-109), as follows:

“All animals were fed up to 24 h before slaughter and then fasted to ensure consistent gastrointestinal conditions, adhering to standard pre-slaughter protocols.”

Comment 5:

2.1 Total number of animals slaughtered? Please specify.

Response 5:

We thank the reviewer for the clarification. The number of animals is now added (Line 99).

Comment 6:

2.1 please specify the sex of animals. All males or mixed?

Response 6:

We thank the reviewer for the clarification. The gender of animals is now added (Line 99). We used only bulls to exclude sex factor in our assessment.

Comment 7:

Line 85: What time post-mortem animals were deboned? Please include

Response 7:

We thank the reviewer for the clarification. We conducted postmortem processing after 24 h of chilling (as a standard regulation for beef carcasses) to mitigate the effects of rigor mortis. This information has been added to the methods section (Lines 128-130).

Comment 8:

Line 97: Specifications of vacuum packaging material and equipment?

Response 8:

We thank the reviewer for the clarification. We have now added information about the packaging material (Lines 148-151).

Comment 9:

Line 97: Specifications of refrigerator?

Response 9:

We thank the reviewer for the clarification. We now added the specification of the refrigerator (Lines 151-152).

Comment 10:

Line 99: What do you mean by three times repeated measurements? Experiment was independently repeated three times? Or three recordings were taken?

Response 10:

We thank the reviewer for the clarification. To ensure accuracy and reliability in color measurements, the Konica colorimeter was set to average three readings per sample. Additionally, we repeated this process three times for each sample and used the average of these results as the final measurement. This procedure has been detailed in the relevant section (Lines 163-173).

Comment 11:

Please include a table indicating treatments.

Response 11:

We thank the reviewer for the suggestion. We prepared a detailed Figure about the research procedure (Figure 1), as follows:

Figure 1. Schematic diagram of research.

Comment 12:

Line 114: "Moreover, the chroma value (C*) and hue angle (hº) were calculated according to the formulas as follows". Please give citation here.

Response 12:

We thank the reviewer for the suggestion. The citation is now added (Line 173).

Comment 13:

2.4 Water holding capacity: Please provide citation.

Response 13:

We thank the reviewer for the suggestion. The citations are now added (Line 178 and Line 182).

Comment 14:

2.8 Sensory analysis: How much sensory sessions were conducted? Or all samples were tested in a single session? In that case, how much total time was taken to conduct this study? It was a single sitting testing? Please specify.

Response 14:

We thank the reviewer for the suggestion. Considering the high numbers of our subsamples, we divided the testing into sessions. We added a part for the clarification (Lines 241 and Line 247), as follows:

Sensory evaluation was conducted separately for the two muscle types across four sessions. In the first two sessions, panelists were presented with two sub-samples from each breed, totaling 8 sub-samples per session. To prevent order effects and bias, the presentation of samples was randomized in each session. Each session lasted approximately 1 h, including brief breaks between samples to cleanse the palate and reset sensory receptors. This setup was specifically designed to prevent fatigue and ensure sharp sensory perception throughout the assessment.”

Comment 15:

Sensory analysis: For general acceptance and overall liking, the panel should be large enough to reach to a conclusion. I will recommend to delete these.

Response 15:

We thank the reviewer for the clarification. These ratings do not reflect the general acceptance among the panelists; instead, each panelist was asked a separate question. This distinction was clarified to the panelists before each session. “General acceptance” refers to how much they liked the effect of the marinade, while “overall like” indicates whether they would purchase the meat again if encountered in a market or restaurant. We included this additional parameter specifically to gather insights on potential market or restaurant preferences (Lines 234-236). We also added the information to Table 5 and Table 6 to prevent misunderstandings.

Comment 16:

Why the interactions of muscle types were studied?

Response 16:

We thank the reviewer for the question. The muscles studied, m. longissimus thoracis et lumborum and m. semimembranosus (topside), differ anatomically in location, fat content, and fiber structure, leading to distinct meat quality characteristics. Therefore, rather than incorporating muscle type as a half factor in our statistical model, we treated the two muscles separately to enhance the precision and reliability of our findings. We analyzed marination time, treatment, and breed interactions for each muscle type individually. This approach, we believe, is more suited for these muscles, which are evaluated differently in markets and restaurants.

Comment 17:

Abbreviate WHC, WBSF, and CL when first time these appeared in the manuscript other than abstract. Please check it.

Response 17:

We thank the reviewer for the correction. We revised the abbreviations as suggested (Lines 94-95 and Line 203).

Comment 18:

Lines 14-15: Please italicize muscle names here and check these throughout the manuscript.

Response 18:

We thank the reviewer for the correction. We have rechecked all mentions of muscle types throughout the entire manuscript.

Comment 19:

Line 18: Please correct " cook loss" as "cooking loss".

Response 19:

We thank the reviewer for the correction. We revised as suggested (Line 22).

Comment 20:

Line 20: Please rephrase sentence. Do not use "We". Rephrase

Response 20:

We thank the reviewer for the suggestion. We revised the sentence as suggested (Line 27).

Comment 21:

Line 39: Correct "marinating" with "marination"

Response 21:

We thank the reviewer for the correction. We revised as suggested (Line 43).

Comment 22:

Line 94: Please write author name or first author et al when to mention such like according to, etc. Please correct this throughout the manuscript.

Response 22:

We thank the reviewer for the correction. We revised as suggested (Lines 141-142).

Reviewer 2 Report

Comments and Suggestions for Authors

The manuscript proposal is interesting at the beginning, it develops an adequate introduction and sets out an objective in accordance with the nature of the research. However, in the materials and methods section it shows an inconsistency. They refer to the inclusion in the study of four bovine breeds from a total of 1200 animals, but they do not specify how many bovines correspond to each breed. They also mention the inclusion of 96 samples of loin and 96 of topside, but they do not specify how many samples correspond to each racial group. Although they include the variable in the experimental design and in the results tables they mention the racial effect in the response variables, their research lacks this support. Unless they indicate which was the procedure where the racial effect was included in the study.

Author Response

Comment 1:

The manuscript proposal is interesting at the beginning, it develops an adequate introduction and sets out an objective in accordance with the nature of the research. However, in the materials and methods section it shows an inconsistency. They refer to the inclusion in the study of four bovine breeds from a total of 1200 animals, but they do not specify how many bovines correspond to each breed. They also mention the inclusion of 96 samples of loin and 96 of topside, but they do not specify how many samples correspond to each racial group. Although they include the variable in the experimental design and in the results tables, they mention the racial effect in the response variables, their research lacks this support. Unless they indicate which was the procedure where the racial effect was included in the study.

Response 1:

First of all, we would like to thank the referee for their positive feedback and for highlighting an important point. One of the most critical aspects of meat quality studies is the selection of animal material. Meat quality parameters can vary significantly depending on factors such as age and body weight. Numerous studies have demonstrated that, to compare meat quality parameters reliably, the other characteristics of the cattle being studied should be uniform. Additionally, nutrition and slaughterhouse conditions should be consistent across all subjects.

In this study, we selected animals of the same age (466.30±3.60 days) and slaughter weight (605.25±1.25 kg) from four different breeds. Achieving uniformity in age, body weight, and condition across four distinct breeds requires a large initial population. Therefore, we selected a beef cattle farm with over 1,200 cattle. Experimental bulls were then chosen from this initial population based on data analysis. Samples were collected from a total of 48 bulls, with 12 animals representing each breed.

As shown in Tables 1 and 2, novel relationships were detected for both individual factors and the two-way and three-way interactions of breed with time and treatment. These findings represent the first results of their kind reported in the literature. Relevant explanations have been added to the methods section (Line 99, Line 109, and Lines 118-120).

Reviewer 3 Report

Comments and Suggestions for Authors

Dear Authors,

I was pleased to accept the invitation to review your manuscript. The topic is very interesting and has significant implications for both the industry and consumers.

I have the following suggestions and questions that I would appreciate you addressing:

•             Materials and Methods.

- What was the resting time of the animals in the slaughterhouse pens between arrival and slaughter?

- Was electrical stimulation of the carcasses performed? If so, under what conditions?

- Was the pH evaluated 24 hours post-mortem? This should be indicated.

- What was the duration of the refrigeration period before the marination procedures were initiated?

- Was this maturation period the same for all carcasses in the study?

- How many animals of each breed were involved in the study?

- What was the ratio of marinade to meat in each sample?

- What method was used for pH evaluation—penetration electrode or evaluation in solution?

- What electrode was used for the pH evaluation?

- In the sensory analysis, what was the cooking time for each sample? Was the internal temperature of each sample controlled?

- Cooking loss: Since the treatment was performed at a constant time, it is important that the samples were of equal dimensions. Please indicate the average weight and variation range of the various samples.

- Shear force: Why wasn't it evaluated at 0 hours?

- Shear force: Line 144 – Cylinders with an area of 10x10 cm? Please explain.

- Shear force: You mention that three cylinders were taken from each sample. In total, how many repetitions were carried out for each LD and SM sample?

- Shear force: When evaluating WBSF, the peak force value is usually presented. Why did you not opt for this value?

- Shear force: Since the WBSF parameters presented are uncommon, I suggest including a figure illustrating how they are calculated, with the start and end points for calculating slope and area. The units should be in SI.

·         Results

- When you refer to 0h of marinating time, was the evaluation conducted immediately before the sample came into contact with the marinade? This should be described in the Materials and Methods section.

- **Table 3:** The pH at 0h is recorded as 6.45. Is this value indicative of DFD (Dark, Firm, Dry) meat, or was it measured immediately after the animal's death?

- If this value was measured after rigor mortis, what was the reason for selecting DFD muscles?

- **Table 4:** Why were LD (Longissimus dorsi) samples evaluated at 12, 24, and 72 hours, while SM (Semimembranosus) samples were only evaluated at 12 and 24 hours?

- **Tables 5 and 6:** Sensory evaluations were only conducted at 12 and 24 hours. With such a narrow time interval, were you expecting significant differences? Why did you not evaluate at 72 hours, which might better align with consumer evaluations?

- **Table 7:** Again, regarding 0h of marinating time, the time between death and the determination of this non-marinated value should be indicated in the Materials and Methods section.

- **Table 7:** What is the reason for the pH of non-marinated samples dropping below the minimum pH corresponding to the isoelectric point of muscle proteins?

- Please explain why there was a significant drop in pH between 0 and 12 hours in the non-marinated samples, followed by a subsequent rise.

- The acidity of the marinades could explain much of the pH evolution in the meat. Do you agree? I suggest including the pH value of each marinade in the Materials and Methods section.

- **Table 7:** What is the reason for the pH of the samples marinated in milk (neutral pH) and olive oil to drop below the pH of the isoelectric point of muscle proteins?

- What was the pH evolution in the SM muscle? I could not find the corresponding table.

- **Table 8:** Similar to the pH results, it is not clear when the 0h post-mortem evaluation was conducted. Could you clarify?

- **Table 8:** Please explain the "V-shaped" evolution of WBSF (Warner-Bratzler Shear Force) with a minimum at 12 hours. What biochemical systems caused the hardening of the samples?

- **Table 11:** What parameter is presented?

The conclusions presented in the manuscript are currently written in a general and somewhat self-congratulatory manner. For a scientific article, it is essential to provide objective and specific conclusions that clearly reflect the results and their implications.

I hope these suggestions are helpful.

Comments on the Quality of English Language

Although English is not my first language, I did not experience difficulty in reading the manuscript.

Author Response

General comment:

Dear Authors,

I was pleased to accept the invitation to review your manuscript. The topic is very interesting and has significant implications for both the industry and consumers.

I have the following suggestions and questions that I would appreciate you addressing.

Response:

We would like to thank the reviewer for their positive comments and valuable contributions. The reviewer’s suggestions and corrections have significantly improved the technical quality of our article. Additionally, due to the reviewer’s high level of expertise in meat pH, several important clarifications were made in our article. Therefore, we would like to express our sincere gratitude to the reviewer once again.

Comment 1:

What was the resting time of the animals in the slaughterhouse pens between arrival and slaughter?

Response 1:

We thank the reviewer for the clarification. The resting time of the animals in the slaughterhouse pens before slaughter was approximately 8 hours. This period allowed the animals to recover from transport stress and ensure uniformity in the conditions before slaughter. The information is now added (Lines 113-114).

Comment 2:

Was electrical stimulation of the carcasses performed? If so, under what conditions?

Response 2:

We thank the reviewer for the clarification. All of the carcasses were electrically stimulated for 30 seconds at 60 volts and suspended by the Achilles tendons. The information is now added (Lines 115-116).

Comment 3:

Was the pH evaluated 24 hours post-mortem? This should be indicated.

Response 3:

We thank the reviewer for the clarification. Meat pH was performed at 1 h postmortem (designated as 0h-control in the paper) and also 24 h postmortem. The information is now added (Lines 188-190).

Comment 4:

What was the duration of the refrigeration period before the marination procedures were initiated?

Response 4:

We thank the reviewer for the question. We did not perform a special pre-cooling period after 24h of chilling 4°C of the carcasses. Afterwards, samples were marinated at 4°C. The information is now added (Line 116-117).

Comment 5:

Was this maturation period the same for all carcasses in the study?

Response 5:

We thank the reviewer for the question. All of the carcasses were chilled for 24h at 4°C in a ventilated room (Line 116-117).

Comment 6:

How many animals of each breed were involved in the study?

Response 6:

We thank the reviewer for the clarification. We used a total of 48 bulls for the study, with 12 animals selected from each breed. The information is now added (Line 99).

Comment 7:

What was the ratio of marinade to meat in each sample?

Response 7:

We thank the reviewer for the clarification. The marinating process involved immersing slices of muscle in the marinades. The ratio between meat and marinade was fixed at 1:10 (Line 148). This is a commonly used marination procedure as also described by paper by Latoch et al. recently published in Foods journal.

Comment 8:

What method was used for pH evaluation—penetration electrode or evaluation in solution?

Response 8:

We thank the reviewer for the question. The pH was measured using a penetration electrode specifically designed for meat samples. A Testo 205 digital pH meter was used, which provided accurate and repeatable measurements for meat products.

Comment 9:

What electrode was used for the pH evaluation?

Response 9:

We thank the reviewer for the question. The pH was measured using a Testo 205 spare probe (Mfr Part #0650 2051) specifically designed for meat samples. It was calibrated with pH 4.01 and pH 7.00 standard buffer solutions (Testo) and was set to correct pH values automatically, taking into account muscle temperature.

Comment 10:

In the sensory analysis, what was the cooking time for each sample? Was the internal temperature of each sample controlled?

Response 10:

We thank the reviewer for the question. We added the details of the cooking process (Lines 219-223), as follows:

Steaks were individually arranged on mesh racks set within aluminum pans, with two thermocouples embedded at the geometric center of each to monitor the internal temperature. The cooking process lasted approximately 20 min, continuing until the samples reached a final internal temperature of 75°C.”

Comment 11:

Cooking loss: Since the treatment was performed at a constant time, it is important that the samples were of equal dimensions. Please indicate the average weight and variation range of the various samples.

Response 11:

We thank the reviewer for the clarification. We agree with the reviewer that initial sample properties are very crucial for the evaluation of cooking loss. For this reason, we aimed to create fairly uniform starting samples. The information of the initial sample weight is now added to methods section (Lines 195-196), as follows:

All samples, each 150±2.80 g, were uniformly cut to the same shape”.

It is important to note that, initially, we have added the regression effect of the initial sample weight to statistical model. The effect was not statistically significant (because the average was very consistent across each sample), and hence, we excluded this factor to provide higher coefficient of determination in the ultimate statistical model.

Comment 12:

Shear force: Why wasn't it evaluated at 0 hours?

Response 12:

We thank the reviewer for the correction. We of course evaluated the 0h samples because they are control samples. But we forgot to mention it in the methods section for Warner–Bratzler shear force assessment (WBSF). The results have been already presented in Tables 3 and 4. The sentence is now revised (Lines 204-205), as follows:

“Following the corresponding marination time (12h, 24h, and 72h), both marinated and control samples were cooked…”

Comment 13:

Shear force: Line 144 – Cylinders with an area of 10x10 cm? Please explain.

Response 13:

We thank the reviewer for the question. dimensions are 10x10 mm (1x1 cm) cross-sectional area (Line 206), which is standard for Warner-Bratzler shear force testing.

Comment 14:

Shear force: You mention that three cylinders were taken from each sample. In total, how many repetitions were carried out for each LD and SM sample?

Response 14:

We thank the reviewer for the question. For each sample, at least three repetitions (3 to 6) were made for both LD and SM samples.

Comment 15:

Shear force: When evaluating WBSF, the peak force value is usually presented. Why did you not opt for this value?

Response 15:

We thank the reviewer for the question. We agree with the referee’s opinion. To comply with the referee’s valuable suggestions, we reviewed our data and added the WBSF peak force analysis to our study. The results are presented in Tables 1-4.

Comment 16:

Shear force: Since the WBSF parameters presented are uncommon, I suggest including a figure illustrating how they are calculated, with the start and end points for calculating slope and area. The units should be in SI.

Response 16:

We thank the reviewer for the suggestions. We concur with the reviewer’s insights. In response to their valuable suggestions, we reassessed our data and incorporated WBSF peak force analysis into our study. The findings are displayed in Tables 1-4. We also provided a figure for calculating the WBSF values and some examples from analysis. In this context; peak force is the force needed to shear through the meat; slope or shear firmness is the slope between the origin and the peak of the curve, and total work is the total energy needed to shear through the meat. We would like to extend special thanks to the referee for their assistance in addressing this deficiency.

Results

Comment 17:

When you refer to 0h of marinating time, was the evaluation conducted immediately before the sample came into contact with the marinade? This should be described in the Materials and Methods section.

Response 17:

We thank the reviewer for the clarification. In this study, we used values from approximately 1-2 hours after slaughter as well as non-marinated samples 24 hours later. We developed a comprehensive mixed model incorporating these control samples and others under various conditions. However, we acknowledge the reviewer's criticism and have made revisions accordingly. Specifically, ‘0h-control’ is no longer included in the marination timeline in Tables 3 and 4, and we have added relevant explanations in the legends.

Comment 18:

**Table 3:** The pH at 0h is recorded as 6.45. Is this value indicative of DFD (Dark, Firm, Dry) meat, or was it measured immediately after the animal's death?

Response 18:

We thank the reviewer for the clarification. The pH “0h-control” is 1h postmortem. Thus, this is a normal pH value but we agree with the reviewer we had to state this information. The corresponding information is now added to Tables 3 and 4, and also to methods section (Lines 188-190).

Comment 19:

If this value was measured after rigor mortis, what was the reason for selecting DFD muscles?

Response 19:

We thank the reviewer for the clarification. In light of the above explanations, we used the 1-hour postmortem pH measurement as the “0h-control.” Consequently, these beefs are not classified as DFD (dark, firm, and dry). We have included relevant explanations in the paper.

Comment 20:

**Table 4:** Why were LD (Longissimus dorsi) samples evaluated at 12, 24, and 72 hours, while SM (Semimembranosus) samples were only evaluated at 12 and 24 hours?

Response 20:

We thank the reviewer for the clarification. This is one of the limitations of our study. This situation is now stated as a limitation in the last paragraph of the discussion section (Lines 695-698).

Comment 21:

**Tables 5 and 6:** Sensory evaluations were only conducted at 12 and 24 hours. With such a narrow time interval, were you expecting significant differences? Why did you not evaluate at 72 hours, which might better align with consumer evaluations?

Response 21:

We thank the reviewer for the questions. We concur with the reviewer that a 72-hour panel review would have been advantageous. However, we excluded this time point for two primary reasons. We are committed to responding to all your questions with sincerity. First, our panelists, who were not professional tasters, were reluctant to consume meat that had been marinated in milk or lemon for three days. Second, extending the sampling to include this timeline would have reduced the volume of samples we could manage in the initial two periods. Consequently, we omitted the 72-hour timeline from our study.

Comment 22:

**Table 7:** Again, regarding 0h of marinating time, the time between death and the determination of this non-marinated value should be indicated in the Materials and Methods section.

Response 22:

We thank the reviewer for the clarifications. We wish to clarify that the analysis conducted involves a three-way interaction within a statistical model. Consequently, the adjusted averages of the factors are compared based on their proportional effects. In this context, the “0h control” samples refer to pH values measured one-hour post-slaughter. Additionally, non-marinated samples from 24 hours postmortem were also included in this statistical evaluation. Our study comprehensively assesses the process from slaughter through the conclusion of the experimental procedures. Detailed explanations have been added to the methodology section.

Comment 23:

**Table 7:** What is the reason for the pH of non-marinated samples dropping below the minimum pH corresponding to the isoelectric point of muscle proteins?

Comment 23:

We thank the reviewer for the question. We believe that the reason for this is entirely due to acidic solutions.

Comment 24:

Please explain why there was a significant drop in pH between 0 and 12 hours in the non-marinated samples, followed by a subsequent rise.

Response 24:

We thank the reviewer for the question. As we mentioned earlier, the “0h control” samples refer to pH values measured one-hour postmortem. Hence the decrease is an expected drop.

Comment 25:

**Table 7:** What is the reason for the pH of the samples marinated in milk (neutral pH) and olive oil to drop below the pH of the isoelectric point of muscle proteins?

Response 25:

We thank the reviewer for the question. We concur with the referee's observation that significant decreases in pH were observed in our study. However, we attribute these decreases primarily to the marination process. In fact, when comparing different marination methods, the results appear consistent and mutually supportive, reinforcing the logical connection between marination and pH reduction. In addition, Ultra-High Temperature (UHT) milk typically has a slightly acidic pH, usually ranging from 6.7 to 6.8. During UHT processing, small amounts of lactic acid can form, contributing to the milk’s slight acidity. Lactic acid, being a weak acid, has the potential to lower the pH of beef during marination. Although UHT processing inactivates many enzymes in milk, some residual enzymatic activity may persist, potentially breaking down meat proteins and releasing amino acids, which might further contribute to a slight decrease in pH. As the milk penetrates the meat during marination, its slightly acidic nature can gradually influence the overall pH of the beef. Extended marination times would allow for more pronounced pH changes. However, it is important to note that while UHT milk can slightly decrease the pH of beef, this effect is likely mild compared to more acidic marinades containing vinegar or citrus juices.

Comment 26:

What was the pH evolution in the SM muscle? I could not find the corresponding table.

Response 26:

We thank the reviewer for the question. Least squares means and their corresponding standard errors are presented in Table 4.

Comment 27:

**Table 8:** Similar to the pH results, it is not clear when the 0h post-mortem evaluation was conducted. Could you clarify?

Response 27:

We thank the reviewer for the question. Table 8 presents the adjusted means of the effects of the triple interaction between marination time (~1 hour postmortem), treatment, and breed in the analysis of variance. These results were derived from a mixed statistical model that compares marination time points with other factors. pH and color measurements were taken 1 hour after slaughter and repeated 24 hours after chilling. While many studies have focused solely on the effects of marination, our approach attempts to account for changes influenced by all possible factors throughout the process, from slaughterhouse to plate. We kindly request the esteemed referee to consider this comprehensive approach in their evaluation.

Comment 28:

**Table 8:** Please explain the "V-shaped" evolution of WBSF (Warner-Bratzler Shear Force) with a minimum at 12 hours. What biochemical systems caused the hardening of the samples?

Response 28:

We thank the reviewer for the question. The "V-shaped" evolution of WBSF with a minimum at 12 hours, as observed in Table 8, can be explained by the complex biochemical processes that occur during marination. Initially, during the first 12 hours, the marination process likely induces tenderization through the action of proteolytic enzymes, which break down muscle proteins, particularly collagen. This leads to a reduction in shear force as the meat becomes more tender. However, after this initial tenderization phase, a hardening of the samples may occur due to the reformation of protein cross-links or the loss of water-holding capacity. This hardening effect could be attributed to the partial denaturation of proteins or the impact of marination ingredients that cause dehydration or protein aggregation, leading to an increase in WBSF values after 12 hours. Essentially, the decrease in WBSF at 12 hours is due to enzymatic tenderization, while the subsequent increase could be related to the reorganization of the muscle fibers and the effect of marination on protein structure, causing a firmer texture.

Comment 29:

**Table 11:** What parameter is presented?

Response 29:

We thank the reviewer for a critical clarification. The parameter is “overall liking” in panel assessment. We revised the Table 11.

Comment 30:

The conclusions presented in the manuscript are currently written in a general and somewhat self-congratulatory manner. For a scientific article, it is essential to provide objective and specific conclusions that clearly reflect the results and their implications.

Response 30:

We thank the reviewer for the suggestions. The conclusions section is now revised.

Reviewer 4 Report

Comments and Suggestions for Authors

Dear Authors,

My comments are attached.

Comments on the Quality of English Language

English editing is required.

Author Response

General comment:

The manuscript presents a study on beef marination, aiming to explore the effects of different marination treatments, times, and cattle breeds on various meat quality parameters. However, the manuscript needs to improve in several areas, including clarity, organization, depth of analysis, and scientific rigour. The hypothesis is not clearly stated, and the methodology is described vaguely.

The manuscript does not follow journal instructions (e.g., spacing, cited literature), and English needs editing.

Response:

We express our gratitude to the reviewer for their insightful comments and valuable contributions. The reviewer's detailed suggestions and thoughtful corrections have not only enhanced the technical quality of our article but have also provided clearer directions for our analysis and presentation. These improvements have undoubtedly strengthened the overall rigor and readability of our work, making the findings more accessible and impactful for our readers.

We have reviewed and revised the article in accordance with the journal’s guidelines. Additionally, we sought professional assistance from MDPI English Language Editing Services to enhance the English language quality of our manuscript.

Abstract:

Comment 1:

The description of the marinating process needs to be more specific – e.g. specify the proportions or concentrations used.

Response 1:

We thank the reviewer for a critical clarification. We have included the details as suggested (Lines 18-19).

Comment 2:

Please mention in one sentence why these specific marinades were chosen.

Response 2:

We thank the reviewer for the suggestion. We mentioned a sentence about the marinades as suggested.

Comment 3:

The abstract mentions that measurements were taken three times, but it needs to be clarified whether these were repeated measures on the same sample or on separate samples.

Response 3:

We thank the reviewer for the suggestion. We mentioned “for each sample” as suggested (Line 23).

Comment 4:

Lines 20-22: Please rephrase. The abstract mentions significant influences of marination time and interactions but does not provide specific details on how these variables affected each measured parameter. Also, it does not specify which factors were included in the model (e.g., marination time, breed, muscle type).

Response 4:

We thank the reviewer for the suggestion. However, the journal allows only 200 words for the abstract. We have reached the word limit allowed for this section and therefore cannot include additional details.

Comment 5:

Line 22: “We showed” please delete and change (e.g. results showed).

Response 5:

We thank the reviewer for the suggestion. Revised as suggested (Line 27).

Comment 6:

Lines 25-26: The bold statement that this study "represents one of the most extensive analyses conducted on the topic of beef marination" lacks context or comparison.

Response 6:

We thank the reviewer for the suggestion. This sentence is now deleted.

Comment 7:

Lines 26-27: What are the practical implications for the industry?

Response 7:

We thank the reviewer for the question. The journal “Foods” allows only 200 words for the abstract. We have reached the word limit allowed for this section and therefore cannot include additional details. We changed “industry” to “beef industry” (Lines 30-31).

Introduction:

Comment 8:

The introduction should emphasize the main research hypothesis, underlining its significance and the reason for conducting this research. The authors' idea that consumer preferences drive meat quality should be expanded with specific examples or relevant studies, as the current literature review is brief and lacks a deeper discussion of previous findings. The introduction needs more structure. Please strengthen the connections between the points discussed (e.g., how consumer preferences lead to the development of marination techniques).

Response 8:

We thank the reviewer for the suggestions. We revised the section (Lines 57-80).

Comment 9:

Please explain why you used these specific marinades, time and beef parts.

Response 9:

We thank the reviewer for the suggestion. We included a specific part based on the suggestions from the reviewers (Lines 68-79).

Comment 10:

The introduction should provide a stronger justification for why these knowledge gaps in meat marinating are significant and how filling them will contribute to the field.

Response 10:

We thank the reviewer for the suggestion. We included a specific part based on the suggestions from the reviewers (Lines 80-86).

Comment 11:

Lines 66-77: Were four cattle breeds fed the same, slaughtered at similar ages, and slaughtered at similar weights? This claim is very strange. Please provide some details regarding the gender, diet composition, and slaughtering techniques.

Response 11:

We thank the reviewer for the clarifications. Indeed, a significant strength of our study lies in the meticulous selection of our subjects. Age, body weight, and physical condition are critical factors that profoundly influence meat quality parameters. Our research was conducted within a large enterprise known for raising various breeds of beef cattle. From a total of 1,200 cattle, we carefully selected 48 animals that were uniform in age, body condition, and weight. The meat from these selected animals was then used to ensure consistency and reliability in our findings. This methodological rigor adds substantial value to our study, providing robust insights into the impact of these factors on meat quality. As a result, the selected animals were uniformly exposed to identical environmental factors, thereby constituting a population in which breed was the sole variable (Lines 118-120).

We included a specific part based on the suggestions about the gender (Line 99), diet composition (Lines 102-107), and slaughtering practices (Lines 113-117).

Comment 12:

Lines 80-83: It is unclear what this sentence means – how and based on what did you choose the marinating procedure? Based on some chef's recipe???

Response 12:

We thank the reviewer for the clarifications. Citrus marinade was prepared following the method reported by Burke et al., 2003. Olive oil-garlic, a traditional marinade from the Mediterranean and Aegean regions, was formulated based on a recipe provided by a chef from an upscale restaurant and was consistently optimized across all samples. Similarly, in the milk marinade, as with the other marinades, a meat to marinade ratio of 1:10 was maintained. Furthermore the details are now provided based on the suggestions (Lines 140-151).

Reference: Burke, R.; Monahan, F., The tenderisation of shin beef using a citrus juice marinade. Meat Science 2003, 63 (2), 161-168.

Comment 13:

Line 95: pH?

Response 13:

We thank the reviewer for the correction. PH of the marinade is now added (Line 145).

Comment 14:

What was the size/weight of meat samples, as this will strongly influence the marinating process?

Response 14:

We thank the reviewer for the correction. We added the details as suggested (Lines 134-136).

Comment 15:

How did you marinate these samples—in a bowl, infused, in a fridge? Are they mixed/turned during this process?

Response 15:

We thank the reviewer for the clarification. The beef samples marinated for 12h, 24h, and 72h at 4°C in a refrigerator. The individual muscle samples, including non-marinated control samples, were packed in high-density polyethylene (HDPE) refrigerator storage containers and wrapped with commercial-grade polyvinyl chloride film (Sera, Turkiye). Samples were turned approximately every 6 h. Concurrently, the pH of the marinades was measured, and the marinades were replaced with fresh ones to maintain consistent pH levels throughout the marination process. We added the details as suggested (Lines 145-155).

Comment 16:

Line 97: It is written that “the samples were vacuum packed and marinated.” If they are vacuum-packed, how can they be marinated? Is the marinade inside vacuum bags with meat?

Response 16:

We thank the reviewer for the clarification. The reviewer correctly noted an ambiguity concerning the use of a special vacuum system, which we did not employ. This oversight might have led to confusion. Therefore, we have revised this section to clarify (Lines 148-151).

Comment 17:

In the results, there are no 72h data for semimembranosus.

Response 17:

We thank the reviewer for the clarification. This is one of the limitations of our study. This situation is now stated as a limitation in the last paragraph of the discussion section (Lines 695-702).

Results:

Comment 18:

This section needs extensive editing and is hard to follow. Why are some of the statistics in the table 3-factor design, and some are not? Please consult an expert for this part and change it completely.

Response 18:

We thank the reviewer for the questions and interpretations. In our study, the longissimus muscle was evaluated at three time points and the semimembranosus muscle at two. We have acknowledged this as a limitation in our research. In addition, our panelists, who were not professional tasters, were reluctant to consume meat that had been marinated in milk or lemon for three days. Hence, we could only use the panel assessment for two time points (12h and 24h). As a result, evaluations were conducted with three factors in some instances and two factors in others. Furthermore, a researcher from the Food Science Department joined our team during the revision phase, allowing us to collaboratively review and refine the entire methodology, as suggested.

Comment 19:

Some sentences are long and complex, making them difficult to follow.

Response 19:

We enlisted the help of MDPI English Language Editing Services to refine the English quality of our manuscript.

Comment 20:

The conclusion makes several strong claims, such as "the most comprehensive investigation of its kind" and "pioneering." These statements should be toned down or supported by specific evidence or comparisons with other studies.

Response 20:

We thank the reviewer for the interpretations. In response, we have removed the specified statements and revised the conclusions section according to the reviewers’ suggestions.

Comment 21:

The phrase "providing invaluable insights" is vague – please be more specific.

Response 21:

We thank the reviewer for the suggestion. We removed this statement and incorporated the results into the conclusions section.

Comment 22:

The conclusion mentions the "negative effects" of marination but does not elaborate on what these are.

Response 22:

We thank the reviewer for the suggestion. We intended to report the undesirable results from the panel assessment. However, following the reviewer’s interpretation, we have omitted this statement to prevent misunderstandings and made the necessary revisions to the text.

Comment 23:

What were the most important results, and how do they contribute to existing knowledge?

Response 23:

We thank the reviewer for the question. We have completely revised the conclusion section.

Comment 24:

What are the future research directions and the limitations of this study? The suggestion for further “molecular analysis” needs to be expanded.

Response 24:

We thank the reviewer for the suggestion. Meat marination is a dynamic process involving multiple interacting chemical reactions. While many studies, including ours, offer macroscopic and observational insights, there is a need for more mechanistic perspectives to deepen our understanding of these processes. Additionally, the genetic makeup of animals has been shown to significantly affect Warner-Bratzler Shear Force, pH, and meat color. Thus, exploring the interactions between an animal's genetic structure and its response to marination could yield novel and intriguing findings. For example, some major genes such as CAPN1, CAST, and GHR are known to significantly influence meat quality attributes including tenderness, which can affect how meat responds to marination processes. It is important to note that the conclusions section has been completely revised based on reviewer’s suggestions. We have also added a specific paragraph to address the limitations of the study (Lines 695-702).

Round 2

Reviewer 1 Report

Comments and Suggestions for Authors

Comment 14 of review round 1: If there were different sessions, how statistical analysis was conducted for sensory analysis? Sessions were taken as fixed or random effects? Please verify and include it in the statistical model section.

Author Response

Comment 14 of review round 1:

If there were different sessions, how was statistical analysis conducted for sensory analysis? Sessions were taken as fixed or random effects? Please verify and include it in the statistical model section.

Response:

We thank the reviewer for their cautious and professional assessment. Indeed, we provided information on the carcass-related factors excluded from the model for simplification (Lines 262-264). However, we inadvertently omitted this information for the panel. We extend our gratitude once again to the reviewer for their attention to detail.

In this study, we employed a mixed-model approach in ANOVA. Given the ample number of samples, we aimed to select the most appropriate statistical model by collectively evaluating various factors. To achieve the highest degree of coefficient of determination (R2), we assessed sub-factors and determined their inclusion in the final model. In this context, the session factor has already been evaluated. We added the relevant information to the statistical analysis section (Lines 264-266) as follows:

The session factor was added as a random effect to account for potential variability between sessions. Since it was not significant, it was excluded from the final statistical model.

Reviewer 2 Report

Comments and Suggestions for Authors

The authors have made the suggested changes which support their results and conclusions.

Author Response

Reviewer #2:

The authors have made the suggested changes which support their results and conclusions.

Response:

We would like to thank the referee for their valuable contributions.

Reviewer 4 Report

Comments and Suggestions for Authors

Dear Authors,

I have no further comments.

Comments on the Quality of English Language

Minor English editing is required.

Author Response

Reviewer #4:

Dear Authors, I have no further comments.

Response:

We would like to thank the referee for their valuable contributions.
